# Imbalanced Power Spectral Generation for Respiratory Rate and Uncertainty Estimations Based on Photoplethysmography Signal

**DOI:** 10.3390/s25051437

**Published:** 2025-02-26

**Authors:** Soojeong Lee, Mugahed A. Al-antari, Gyanendra Prasad Joshi, Yeong Hyeon Gu

**Affiliations:** 1Department of Computer Engineering, Sejong University, 209 Neungdong-ro, Gwangjin-gu, Seoul 05006, Republic of Korea; 2Department of Artificial Intelligence and Data Science, Sejong University, Seoul 05006, Republic of Korea; en.mualshz@sejong.ac.kr; 3Department of AI Software, Kangwon National University, Samcheok 10587, Kangwon State, Republic of Korea; joshi@kangwon.ac.kr

**Keywords:** photoplethysmography, respiratory rate estimation, uncertainty, power spectral curves, bootstrap

## Abstract

Respiratory rate (RR) changes in the elderly can indicate serious diseases. Thus, accurate estimation of RRs for cardiopulmonary function is essential for home health monitoring systems. However, machine learning (ML) algorithm errors embedded in health monitoring systems can be problematic in medical decision-making because some data have much larger sample sizes in the training set than others. This difference in sample size implies biosignal data imbalance. Therefore, we propose a novel methodology that combines bootstrap-based imbalanced continuous power spectral generation (IPSG) with ML approaches to estimate RRs and uncertainty to address data imbalance. The sample differences between normal breathing (12–20 breaths per minute (brpm)), dyspnea (≥20 brpm), and hypopnea (<8 brpm) show significant data imbalance, which can affect the learning of ML algorithms. Hence, the normal breathing part with a large amount of data is well-trained. In contrast, the dyspnea and hypopnea parts with relatively fewer data are not well-trained, and this data imbalance makes it difficult to estimate the reference variables of the actual dyspnea and hypopnea data parts, thus generating significant errors. Hence, we apply ML models by mixing artificial feature curves generated using a bootstrap model with the original feature curves to estimate RRs and solve this problem. As a result, the nonparametric bootstrap approach significantly increases the number of artificial feature curves. The generated artificial feature curves are selectively utilized in the highly imbalanced parts. Therefore, we confirm that IPSG is efficiently trained to predict the complex nonlinear relationship between the feature vectors obtained from the photoplethysmography signal and the reference RR. The proposed methodology shows more accurate prediction performance and uncertainty. Combining the proposed Gaussian process regression (GPR) with IPSG based on the Beth Israel Deaconess Medical Center dataset, the mean absolute error of the RR is 0.79 and 1.47 brpm. Our approach achieves high stability and accuracy by randomly mixing original and artificial feature curves. The proposed GPR-IPSG model can improve the performance of clinical home-based monitoring systems and design a reliable framework.

## 1. Introduction

As the number of older adults living alone increases, monitoring their respiratory rates (RRs) has become essential. Rapid changes in RRs can be a sign of health problems in older adults [1,2]. Therefore, it is crucial to estimate the RR accurately to manage cardiopulmonary health. In older individuals, rapid changes in RRs suggest serious illness. Tachypnea is a fast, shallow RR in adults, which is usually defined as more than 20 breaths per minute (brpm) while resting. It can indicate an underlying health problem that requires medical attention, especially if other symptoms, such as chest pain, dizziness, or fatigue, accompany it [3]. Hence, accurately estimating RRs for cardiorespiratory health is vital. At present, capnography is the standard method for measuring RRs. However, this method is expensive and requires strict management [4].

However, available non-contact RR measurement systems currently have limitations in representing this uncertainty because they employ a single measurement value. Applying a pulse oximeter with oxygen saturation (SpO2) and heart rate to obtain accurate RRs is both patient-friendly and economical [5]. Biomedical and engineering researchers have used photoplethysmography (PPG) signal-based wearable devices to predict SpO2. Furthermore, PPG and electrocardiography (ECG) sensor-based continuous signal measurement approaches have been employed to estimate RRs [6,7]. Recently, Lee et al. [8] developed a novel method to estimate RRs and confidence intervals (CIs), using the Gaussian process regression (GPR) model. However, the CI obtained from the GPR model for representing uncertainty is computed based on the distribution of RR estimates [8], which provides relatively wide CIs. This wide CI increases the probability that the RR estimate will fall within the relatively wide CI. Still, it has the disadvantage of reducing the reliability of the RR monitoring system. In addition, although GPR is powerful and flexible, its dependence on scalability, computational cost, kernel selection, and hyperparameter tuning may make it impractical for large-scale applications or complex and high-dimensional data. In addition, if the CI is too narrow, even a tiny change in the RR estimate can easily lead to an out-of-CI, which limits its role as an RR monitoring system. Securing an appropriate range of CIs is necessary to overcome these shortcomings.

Over the past decade, machine learning (ML) has been widely applied in biomedical engineering to estimate objective values [8,9,10]. Lee et al. [11] presented an ensemble gradient boosting algorithm (EGBA) with PPG signals for estimating RRs. Furthermore, multiphase feature-based EGBA has improved RR estimation [11]. However, if multiple feature extraction methods are utilized, inappropriate feature vectors may be selected based on the feature extraction method, which diminishes the reliability of RR estimation [11]. Thus, a practical feature extraction method can be incorporated into an ML algorithm in the automatic feature extraction process. An ensemble-based gradient boosting algorithm (GBA) based on multiphase features [11] was applied to improve the performance of RR prediction. In particular, several techniques such as the autoregressive method [12], multiple fractal wavelet readers [13], wavelet packets [14], and maximum overlap discrete wavelet transform [15,16] were used to extract features to compensate for insufficient data. However, EGBA based on multiphase features also had the disadvantage of taking much time to extract features required for RR estimation from PPG signals as it required collecting various features in multiple phases. It was difficult to determine which features should be used and which ones were needed to obtain the optimal estimation rate.

Additionally, computer resource loss may occur when multiple transformation methods are utilized in feature extraction [8]. Liu et al. [17] introduced generative boosting via long short-term memory (LSTM) for estimating RRs. Subsequently, Kumar et al. [18] applied LSTM to estimate RRs using PPG and ECG signals. LSTM is primarily used in time domain and image data processing. However, LSTM has the following limitations: a complex structure, use of several parameters, severe fluctuations in the predictive performance based on the data used, and excessive consumption of computer resources [8]. Support vector machine (SVM) algorithm assumes a kernel that determines the mapping between the feature and target vectors. Then, a training dataset is utilized to estimate the unknown parameters of the kernel function. Singh et al. [19] used to integrate the SVM as a simple structure based on the radial basis function kernel for automatic classification of asthma and non-asthma. Thus, these algorithms are different from parametric algorithms.

ML algorithms trained using limited data, including the neural network (NN) and LSTM models, are generally sensitive to the training sequence and initial parameter values, which results in unstable performance [20]. Therefore, like most ML algorithms, collecting reasonably sized data is vital for training ML models. However, in the biomedical field, data collection encompasses several challenges and practical limitations. First, patient data collection requires significant time and effort. Second, it is often difficult to obtain many patients with a particular disease owing to its low prevalence [21]. Therefore, research on the effects of limited sample size is attracting attention.

Furthermore, ML errors associated with using imbalanced data have garnered less attention, which has primarily been covered in the literature in the field of ML [22,23]. Data imbalance can drastically affect the training of traditional ML algorithms [24]. It involves both convergence during the training step and generalization of an algorithm on the test set [22]. Two representative models augment imbalanced datasets in classification models that use ML algorithms. Under sampling methods, most class samples are randomly removed to balance the class distribution. However, eliminating the majority of class samples can result in information loss and deteriorate the classifier’s performance. Over-sampling methods randomly replicate samples of the minority class to balance the class distribution [25]. Sun et al. [26] presented an over-sampling method that effectively addresses the challenge of negative over-sampling results in order to generate new synthetic samples in the minority class region.  However, over-sampling also has drawbacks such as the possibility of overfitting, introducing noise into the dataset, increasing computational complexity, and limiting information gain. These shortcomings may affect the generalization ability of ML and its performance in the test environment.

Until recently, data imbalance studies have mainly focused on data classification. However, studies have yet to be found, which address the data imbalance problem in order to improve the accuracy of RR estimation and estimate uncertainty. Data imbalance still existed in the feature dataset. For example, the sample difference between hypopnea (<8 brpm), normal breathing (12–20 brpm), and dyspnea (≥20 brpm) indicates significant data imbalance [27], which is a serious problem when using the ML algorithms because data imbalance can affect the learning of the ML algorithms. The normal breathing samples with more data are well-trained. In contrast, the dyspnea and hypopnea samples with relatively little data are not well-trained, and this data imbalance causes significant errors concerning the reference variables in the actual dyspnea and hypopnea data parts.

Beyond these motivations, we introduce a new method, imbalanced power spectral generation (IPSG) using bootstrap [28], to estimate RRs and uncertainty based on the PPG signals in order to solve the problems mentioned above in this study. The bootstrap approach uniquely addresses the challenges of high variance and non-linearity by creating multiple resampled datasets, offering a flexible, non-parametric approach that estimates variability across different subsets of the data. This helps to produce more stable and reliable results, especially in the face of nonlinear relationships and data with high variance [29]. In our research, the non-parametric bootstrap approach [28] generates continuous power spectral artificial curves using the imbalanced continuous power spectral, i.e., dyspnea and hypopnea signals. The generated power spectral artificial curves are then fused with the power spectral curves extracted from the original dyspnea and hypopnea signals and used as training data. As a result, the proposed IPSG method dramatically increases the number of artificial power spectral curves as continuous features. These artificial power spectral curves are selectively used for severely imbalanced feature sets. Increasing the number of artificial power spectral curves for dyspnea and hypopnea samples can make the dataset more balanced for the conventional ML and deep learning algorithms.

This is the first study to use IPSG and six existing ML algorithms to estimate RRs. The concept of the proposed algorithm is that RRs and uncertainty are directly estimated using ML combined with the IPSG algorithm. Compared with [10], this study makes the following contributions (Figure 1 shows a block diagram of the IPSG algorithm for estimating RRs and uncertainty):A new method for estimating RRs and uncertainty was proposed using the IPSG algorithm based on the GPR algorithm.The non-parametric bootstrap approach was utilized to generate artificial feature curves for estimating RRs in order to address severely imbalanced features.The proposed methodology is helpful in clinical settings or diagnostic applications for estimating RRs and uncertainty by providing a CI and low estimated mean absolute error, possibly suggesting more accurate predictive performance and uncertainty.

## 2. Dataset

### 2.1. Collection of PPG Signals

We used two public datasets to evaluate the proposed algorithm. The first dataset used in this study was extracted from the MIMIC-II resource [30] of the Beth Israel Deaconess Medical Center (BIDMC) [31]. It comprises ECG, PPG, and impedance pneumography (IP) respiratory signals acquired from patients aged 19–90 and admitted to an intensive care unit, with 53 recordings at 125 Hz over 8 min. The patients in the dataset were randomly selected from a cohort of individuals hospitalized at the BIDMC in Boston, USA. Reference RR values were derived using two sets of annotations for individual respirations in the IP signal. These datasets were sequentially separated into 80% for the training and 20% for the testing. As shown in Figure 2a, the distribution of RR reference differences between normal breathing (12–20 brpm), dyspnea (≥20 brpm), and hypopnea (<8 brpm) indicated significant data imbalance [27]. We presented an extended distribution of IPSG-based reference RRs on the training dataset, such as the red dash lines in Figure 2b. We also denoted a distribution of the RR reference on a testing dataset in Figure 2c. We also presented an expanded distribution of the IPSG reference RR on the testing dataset, such as the red dash lines in Figure 2d, based on the BIDMC dataset.

The second RRSYNTH dataset [31] consists of 192 wave signals, each of which consists of a record length of 210 s at a sampling frequency of 500 Hz. The RRSYNTH dataset is an artificial synthetic PPG signal modulated and generated in three ways to reflect the characteristics of various cardiovascular and respiratory diseases. First, baseline modulation (BW) represents the baseline change in the PPG signal well. Second, amplitude modulation (AM) represents the amplitude change in the signal. Third, the interval of one cycle of the PPG signal waveform varies depending on the frequency modulation (FM). Therefore, a total of 192 records of three modulation signals, BW (64 records), FM (64 records), and AM (64 records), were used to test various respiratory diseases. The respiratory signal’s physiological mechanisms are described in [31]. PPG BW modulation is known to be caused by changes in blood volume in the tissue. Blood volume changes are caused by changes in thoracic pressure delivered through the arteries and vasoconstriction of the arteries that carry blood to the veins during inspiration. AM indicates decreased stroke volume during inspiration due to changes in intrathoracic pressure. FM also indicates a spontaneous increase in HR and a decrease in expiration during inspiration, a frequent signal phenomenon when it is a symptom of respiratory arrhythmia [32].

### 2.2. Preprocessing Steps

In biomedical and related research fields, PPG has become a popular tool for estimating RRs, heart rate, and SpO2 and predicting blood pressure. Notably, it has revolutionized our understanding and the monitoring of various physiological parameters in the body. PPG signals were collected using the MIMIC-II and RRSYNTH datasets mentioned in the previous subsection. Next, high-frequency signals were removed using a Kaiser window with a cutoff frequency of 35 Hz and signal bandwidth of 3 dB. A merge segmentation technique was used to segment the PPG signal into multiple pulses, as described in [33]. The reference points were obtained from the highest and lowest points of the PPG signal (Figure 1b). Similarly, the lowest points were obtained. Next, linear interpolation was used to resample the PPG signal at 5 Hz, and low-frequency PPG signals were removed using a low-pass filter (Kaiser window function) with a cutoff frequency of 0.0665 Hz and bandwidth of 3 dB (Figure 1b). Finally, the PPG signals yielded a resampled waveform. The resampled waveform dataset was obtained from individual records or participants. The mean seconds between continuous breathing in the Hamming window were used to estimate the RR values. Using a window size of 32 s [33], our proposed method exhibits better performance than the previously used window sizes of 16 and 64 s. The signal quality index (SQI) we used is intrinsically linked to the PPG peak detector because we use SQI to extract bad signals. We used a three-point peak detector with empirically determined thresholds for PPG pulse-peak detection. For more details, see the references [34].

## 3. Bootstrap-Based Imbalanced Feature Generation

Generative adversarial networks are powerful and versatile tools, especially useful for generating high-quality synthetic data and demonstrating creativity in various fields. However, training stability, expensive computational costs, and evaluation challenges can hinder our practical use. On the other hand, the main advantage of the bootstrap method is that it can provide robust, low computational cost, a simple algorithm, and flexible and reliable estimates of various statistics, especially when the underlying distribution is unknown or the sample size is small [35].

### 3.1. Handling High Variance and Dealing with Non-Linearity Using Bootstrap

Bootstrap is a powerful resampling technique that addresses the challenges of high variance and non-linearity in datasets in several unique ways, which generates multiple resampled datasets by randomly sampling data points from the original power spectral curves with replacement. Each resampled dataset (called a bootstrap sample) has the same size as the original power spectral curves but may contain duplicate or missing data points [28]. Generating many bootstrap samples and analyzing the variability across the results allows for estimating the sampling distribution. This process helps reduce the impact of high variance, as it accounts for how the model might behave across various subsets of the data rather than relying on a single sample [29].

Bootstrap is a non-parametric method, meaning it does not make any assumptions about the functional form of the relationship between variables, which is especially useful for datasets with non-linear relationships. Instead of fitting the data to a specific linear regression, bootstrap allows for exploring the data’s inherent structure by drawing resamples and evaluating performance on each. This provides a built-in method for cross-validation without requiring additional validation sets, which is particularly beneficial in high-variance scenarios where overfitting the training set might otherwise be a concern [29]. Therefore, the bootstrap method creates multiple independent bootstrap power spectral (PS) curves by resampling the original PS curves.

### 3.2. Review of Power Spectral (PS) Feature Extraction

The RR estimation is calculated by multiplying the respiratory rate by 4 for a period of 15 s to convert it to brpm as described in [18]. Using autocorrelation functions, the respiratory frequencies can be obtained as automated feature vectors from the PS. PS features can be automatically extracted from the resampled signals utilizing the autocorrelation function, as shown in Figure 3a, following a preprocessing step detailed earlier. The autocorrelation function measures how similar a signal is to a delayed version of itself over successive time intervals. It is represented utilizing the variance, mean, and covariance. The average value from the input dataset is denoted by $x={\left\{x_{n}\right\}}_{n=1}^{N}$, specifying the expected value E[x] at each discrete time *n*. The mean function is represented by $\mu_{x}=E\left[x\right]$. The autocorrelation function measures the difference between discrete times *n* and n+m. When m=0 (delay 0), the autocorrelation function indicates the maximum value, representing the total energy, and is expressed as follows:(1)$$\nu_{m}\left(x\right)=\frac{\sum_{n=1}^{N}\left(x_{n}-\mu_{x}\right)\left(x_{n+m}-\mu_{x}\right)}{\sum_{n=1}^{N}{\left(x_{n}-\mu_{x}\right)}^{2}}$$

The PS represents the correlation structure of the wave signal obtained by fast Fourier transforming the correlation coefficient [36]. The PS in the time domain exhibits autocorrelation and has the same mathematical properties as the square of the amplitude spectral. This study used a sampling frequency of 125 Hz and a record length of 400 s to acquire PS for all components within the range. Conversion to a PS resulted in 50,000 signal samples for each participant. The PS curves were constructed using the autocorrelation to obtain 12 × 32 windows, and 12 × 257 (=3084) data points were acquired from the signal (400 s) as shown in Figure 3b. This process was repeated for the 53 subjects, resulting in 12 × 53 (=636) samples and × 257 feature dimensions. We included an example of the effect of PS feature curves extraction as shown in Figure 3b.

After signal preprocessing, a clean wave signal was obtained (Figure 1c). This clean waveform was split into segments with a 32-s window. Notably, the 32-s window segments did not overlap. The segmented waveform was converted into an autocorrelation coefficient during data analysis, and the PS was obtained from this coefficient (Figure 3b). The PS is a mathematical tool used to analyze biological signals over time. A fast Fourier transform of the autocorrelation function should be performed to obtain the PS of all the components within 257 samples. After that, the proposed model extracts relevant features from this PS, learns them automatically, and uses them to estimate RRs [36].

### 3.3. Imbalanced Feature Generation Using Bootstrap Power Spectral Curve

Efron originally introduced the bootstrap approach in [28]. The advantage of this bootstrap method is that it does not require any modeling or assumptions regarding the data [37]. In the present study, the bootstrap method was used to generate artificial power curves to increase the number of imbalanced power curves.(2)$$\begin{array}{c} P=\left[\begin{array}{ccc} p_{1,1} & \cdots & p_{1,c}\\ \vdots & \ddots & \vdots \\ p_{n,1} & \cdots & p_{n,c} \end{array}\right] \end{array}$$
where P denotes a PS curves measurement matrix obtained using the PS feature vector, as in (Equation 2), based on the PPG signal. *n* is the number of measurements, and *c* (=257) denotes the feature dimension. From the original PS curves matrix, as shown in Figure 4a and Figure 5a, *B* resample PS matrices were obtained $\left\{P_{1}^{*},P_{2}^{*},...,P_{B}^{*}\right\}$ using the non-parametric bootstrap approach as follows:(3)$$\begin{array}{c} P_{i}^{*}=\left[\begin{array}{ccc} p_{1,1}^{*i} & \cdots & p_{1,c}^{*i}\\ \vdots & \ddots & \vdots \\ p_{n,1}^{*i} & \cdots & p_{n,c}^{*i} \end{array}\right] \end{array}$$
where $P_{i}^{*}$ denotes in the *i*th artificial PS matrix acquired as shown in Figure 4b from the original PS matrix (Equation 2) using the non-parametric bootstrap approach with random replacement of PS feature vectors.(4)$$p_{\mu }^{*}\left(i\right)=\frac{1}{n}\sum_{j=1}^{n}P_{i}^{*}$$
where $p_{\mu }^{*}\left(i\right)$ denotes *i*th bootstrap estimate of the mean obtained from (Equation 3). As a result, we obtain $\left\{p_{\mu }^{*}\left(1\right),p_{\mu }^{*}\left(2\right),\ldots ,p_{\mu }^{*}\left(B\right)\right\}$, where *B* (=100) denotes the number of bootstrap resamples. Algorithm 1 provides a detailed explanation of the procedures for imbalanced feature generation and CI bands. Figure 5b,d show that the mean PS curve of the original feature and artificial feature vectors using the bootstrap approach is almost identical. Figure 5e shows the difference between (b) and (d). In addition, the red line in Figure 5f shows the standard error (SE) of the PS curves, and the blue line shows the SE of the artificial PS curves using the bootstrap. This suggests that the artificial PS curves generated using the bootstrap approach are a reliable approximation of the original power curves.
**Algorithm 1** Imblanced Power Spectral Generation: (IPSG)**Procedure**: IPSG (P): PS curves matrix
     $p_{\mu }$ = mean(P): sample estimate of mean     $p_{s}$ = std(P): sample estimate of standard deviation     $p_{se}$ = $\frac{p_{s}}{\sqrt{(}n)}$: sample estimate of standard error     df=n−1: freedom of sample     $c_{t}$ = tinv(1−α/2,df): $c_{t}$ is estimate of the critical t-value     ${\mathrm{ci}}_{\alpha }=p_{\mu }-c_{t}\cdot p_{se}$: estimate of lower CI     ${\mathrm{ci}}_{\beta }=p_{\mu }+c_{t}\cdot p_{se}$: estimate of upper CI**for **i=1, B** do**     index = randi(n,n,1): random resample function with replacement for PS     $P_{i}^{*}$ = boots(index,:): *i*th bootstrap sample of the PS feature matrix     $p_{\mu }^{*}\left(i\right)$ = mean($P_{i}^{*}$): *i*th bootstrap estimate of the feature’s mean**end for**     where B is the number of bootstrap sample and n is the number of     the distribution of RR reference value (ex: < 8 brpm)     $p_{s}^{*}$ = std($p_{\mu }^{*}$): standard deviation of the bootstrap replications**for **i=1, B** do**     t(i) = max(abs($p_{\mu }^{*}\left(i\right)-p_{\mu }$)/$p_{s}^{*}$): t-statistic**end for**     $c^{*}$ = prctile(t, 100·(1−α/2)): compute test statistic at α(=0.05) level     $p_{\mu }^{*}=\frac{1}{B}\sum_{i=1}^{B}p_{\mu }^{*}\left(i\right)$:     bias = $p_{\mu }-p_{\mu }^{*}$     ${\mathrm{ci}}_{\alpha }^{*}=p_{\mu }^{*}-c^{*}\cdot p_{s}^{*}$: estimate of lower CI     ${\mathrm{ci}}_{\beta }^{*}=p_{\mu }^{*}+c^{*}\cdot p_{s}^{*}$: estimate of upper CI**End procedure**

### 3.4. Kolmogorov–Smirnov Test for Artificial Power Spectral Curve

The two-sample Kolmogorov–Smirnov test was used to return the test results for the null hypothesis that the data of the vectors $f_{1}$ and $f_{2}$ are extracted from the same continuous distribution, where $f_{1}$ is the mean vector for the original sample data below 8 brpm, and $f_{2}$ is the mean vector for the artificial sample data below 8 brpm. The alternative hypothesis was that $f_{1}$ and $f_{2}$ can be extracted from various continuous distributions. If the test rejected the null hypothesis at the 5% significance level, the result *h* was 1; otherwise, it was 0. Figure 6a,b demonstrate the cumulative distribution function (CDF) plots of the mean of the sample data (≤7) brpm and that of the sample data (=10) brpm, respectively. The blue line denotes the mean vector of the original vector, whereas the red dashed line represents the mean vector of the artificial vector. Here, we obtained the results (h=0), p=0.988, and ks=0.016 as shown in Figure 6a. Therefore, we can accept the null hypothesis at the 0.05 significance level. Moreover, we confirmed that the *p*-values of the KS test were more than the 0.05 significance level. Furthermore, we obtained the results (h=0), p=0.991, and ks=0.023, as shown in Figure 6b. Therefore, we can accept the null hypothesis at the 0.05 significance level.

## 4. ML Algorithms

Six benchmark models based on ML and deep learning algorithms to demonstrate the effectiveness of the proposed IPSG model are represented as follows.

### 4.1. SVM

One of the most popular machine learning algorithms these days is SVM [19]. It is a supervised learning algorithm that penalizes high and low mispredictions, allowing us to train SVM using a symmetric loss function. The essence of SVM is to map data to a high-dimensional feature space using nonlinear relationships and then perform linear regression in the space.

### 4.2. NN

Neural networks (NNs) aim to understand the relationship between independent and dependent variables by mimicking the behavioral processes of the human brain. They consist of a set of interconnected artificial neurons organized in a hierarchical structure, which is designed to effectively process various types of data [38].

### 4.3. GBA

The GBA is a powerful and flexible ML algorithm that builds a strong predictive model by iteratively correcting errors in previous models [39]. However, it requires careful tuning to prevent overfitting and ensure good generalization to unseen data. Consequently, GBA can be efficiently trained to predict the complex nonlinear relationship between PPG signals’ fused power spectral curves and the reference RR.

### 4.4. LSTM

Recurrent neural networks are prominent deep learning methods for time series and sequential data. The LSTM network is an extension of simple recurrent neural networks and has been significant in modeling temporal sequences [18].

### 4.5. Random Forest

The random forest (RF) algorithm enhances the bagging method by utilizing both bagging and feature randomness to create a collection of uncorrelated decision trees. Feature randomness, also known as feature bagging, helps reduce the correlation between decision trees by generating random subsets of features. It is the key distinction between decision trees and random forests. While decision trees evaluate all possible feature splits, random forests only consider a subset of the features [40,41].

### 4.6. GPR

GPR [42] is an ML model categorized as a non-parametric model. The GPR model does not utilize specific parameters for input-output mapping and treats it as an arbitrary function with a probability density function based on an a priori Gaussian process (GP). Recently, Lee et al. [43] proposed a novel method for estimating RRs and CIs using the GPR based on the PPG signal.

## 5. Experimental Results

The proposed method, IPSG, was applied to the conventional ML methods to adjust the parameters using the learning step. As mentioned in the introduction, ML models such as SVM [19], NN [44], GBA [11], LSTM [18], RF [40], and GPR [42] were used to compare the performance with the proposed IPSG and have been popular in continuous RR estimation until recently. The SVM model [19] was suitable for handling nonlinear relationships between features extracted from PPG signals and reference RRs as it can handle nonlinear relationships in input and output data well. Many researchers have used the NN models for medical application [20,44]. Different deep learning models have recently been used for RR estimation based on PPG signals [17,18].

The tuning stage of parameters was crucial because it can considerably improve the performance of ML models. This study’s five-fold cross-validation fine-tuned the parameters for the conventional six ML models. The main parameters for each model were defined, and the possible range of values for each parameter was determined automatically using the optimized hyperparameters in MATLAB, highlighting our research’s potential benefits and advancements. The process began with conventional ML models like GBA and LSTM, which conducted a grid search for all possible parameter combinations to identify the parameters for achieving the best performance results. Subsequently, this five-fold cross-validation was performed to enhance the robustness of the ML models using the optimal parameters identified. The training data were then divided into five non-overlapping subsets of equal sizes for iterative learning. Each iteration involved four-fold training and assessment of the model using the remaining folds.

After preprocessing, the resampled wave signals were generated to remove noise and outliers from the PPG signals. The proposed IPSG method was generated using the 53 and 192-long resampled wave signals, respectively. In the first experimental scenario, the PPG signals were sequentially split into 80% and 20% for the training dataset to avoid mixing training and test data. Subsequently, the reference RR values were calculated from the oral and nasal pressure signals using a custom respiration detection algorithm [31]. In the second experimental scenario, the PPG signals were sequentially divided into a training set (70%) and a test set (30%). Overall, the results show a slight increase in mean absolute error (MAE) and root mean square error (RMSE) compared to the first scenario experiment’s results. Table 1 presents the parameter adjustments for the conventional and proposed IPSG models using PPG signals based on the RRSYNTH dataset [31]. Parameter tuning was omitted when using the BIDMC dataset to save paper space since the RRSYNTH dataset is more extensive. MATLAB 2023 [45] is used to measure the execution (training and testing) time, based on the RRSYNTH dataset listed in Table 2. The test time measurement results show that the computational complexity of each benchmark model and IPSG combination is almost the same, so it can be expected that there will be no problem in applying it to a real home-based monitoring system, as shown in Table 2.

Using the BIDMC dataset, we obtained 12 × 53 (=636) samples with 257 feature dimensions, where 53 represents the number of patients. Furthermore, we obtained 100 samples from (7≤), 100 samples from (8 to 9), 100 samples from (10 to 11), 100 samples from (22 to 23), and 100 samples (24 to 25), and we acquired 100 samples from (25≥ brpm), using 636 original samples. We then used them as input data for the proposed IPSG algorithm and acquired 600 artificial samples, using the proposed IPSG method as denoted in Figure 4b. Hence, we acquired 1236 samples with 257 feature dimensions. We also obtained 6 × 192 (=1152) samples with 160 feature dimensions, where 192 denotes the number of records, and combined 1152 samples with 600 artificial samples, using the proposed IPSG method. Hence, we acquired 1752 samples based on the RRSYNTH dataset.

Due to space limitations in the paper, the results of the second experimental scenario are denoted in Table 3, Table 4, Table 5 and Table 6. We will skip the detailed analysis and the discussion. Table 3 denotes the performance evaluation results using the IPSG algorithm for six benchmark algorithms: SVM [19], NN [44], GBA [11], LSTM [18], RF [41], and GPR [42]. For the objective performance evaluation of the IPSG algorithm in this table, the results obtained by combining IPSG with all learning algorithms and without combining the IPSG algorithm were evaluated using the public BIDMC and RRSYNTH datasets. The evaluation used MAE and standard deviation (SD) using PPG-based continuous PS curves. Each continuous PS curve was inputted into each ML model, and the resulting MAE and SD values were recorded.

The average results of 30 iterations show that the SVM-IPSG algorithm (MAE 2.15 brpm) does not outperform the SVM algorithm (MAE 1.98 brpm). Here, the MAE difference between the SVM and the combined SVM-IPSG models is shown in Table 3, which indicates that the MAE increased by 0.17 brpm. The LSTM-IPSG and LSTM algorithms show MAE values of (2.22 brpm and 2.15 brpm), respectively, showing that the combined LSTM-IPSG method performs slightly worse than the LSTM method. However, in Table 3, the original NN and NN-IPSG algorithms denote MAE values of (2.15 brpm and 1.57 brpm), respectively, showing that the proposed IPSG combined NN algorithm performs better. The GBA-IPSG algorithm (MAE 1.17 brpm) outperforms the GBA algorithm (MAE 1.84 brpm). The combined RF-IPSG algorithm (MAE 0.98 brpm) also outperforms the RF algorithm (MAE 1.53 brpm). The GPR-IPSG algorithm achieves better results (MAE 0.79 brpm) than GPR (MAE 1.47 brpm) based on the BIDMC dataset. Table 3 shows that most of the six benchmark algorithms, including SVM, also show robust results in the second experimental scenario.

Using the RRSYNTH dataset, the SVM-IPSG algorithm performed better (MAE 3.66 brpm) than the SVM algorithm (MAE 4.95 brpm). The NN-IPSG and the NN algorithm showed MAE values (3.02 brpm and 3.78 brpm), respectively, indicating that the proposed IPSG combined NN algorithm also showed better performance. The GBA-IPSG algorithm performed better (MAE 3.98 brpm) than the GBA algorithm (MAE 5.03 brpm). The LSTM-IPSG and LSTM algorithms also obtained MAE values of (4.11 brpm and 4.85 brpm), respectively, showing that the LSTM-IPSG combined method represented slightly better performance. The RF-IPSG algorithm (MAE 2.60 brpm) outperforms the RF algorithm (MAE 3.60 brpm). The GPR-IPSG algorithm (MAE 1.47 brpm) obtained better results than the GPR algorithm (MAE 2.24 brpm), as shown in Table 3.

Based on the results presented in Table 4, both the conventional and proposed algorithms are evaluated based on the RMSE and SD values. The SVM-IPSG model had a lower RMSE value (1.88 brpm) than the SVM model (2.29 brpm), which indicated that the RMSE was reduced by 0.41 brpm. Hence, a reduction of 0.41 brpm means that the performance improvement of the RR is simply 17.3% = (2.29 − 1.88)/1.88 × 100 with respect to the performance of the home-based healthcare system. Similarly, the NN-IPSG model had a lower RMSE value (2.57 brpm) than the NN model (2.32 brpm), which shows that the RMSE was reduced by 0.25 brpm. Thus, a reduction of 0.25 brpm means that the performance improvement of the RR is simply 10.8% = (2.57 − 2.32)/2.32 × 100 with respect to the performance of the home-based healthcare system.

The GBA-IPSG model outperformed the conventional GBA model, with RMSE values of 1.73 brpm and 2.50 brpm, respectively. This difference indicates that the RMSE was reduced by 0.77 brpm. Therefore, a reduction of 0.77 brpm means that the performance improvement of the RR is simply 44.5% = (2.50 − 1.73)/1.73 × 100 with respect to the performance of the healthcare monitoring system. As shown in Table 4, the GBA model has a small number of parameters, shows stable results even with little parameter tuning, and shows robust results in the tuning range of the test data ratio from 20% to 30%.

Combining the LSTM-IPSG algorithms obtained a similar RMSE value (2.47 brpm) as the LSTM algorithm (2.54 brpm). The difference between these two algorithms denotes 0.07 brpm, which is a 2.8% = (2.54 − 2.47)/2.47 × 100 performance improvement in terms of the healthcare monitoring system. The RMSE value of the RF-IPSG algorithm (1.69 brpm) was lower than that of the RF model (2.15 brpm), which shows that the RMSE was reduced by 0.46 brpm. Therefore, a reduction of 0.46 brpm means that the performance improvement of the RR is simply 27.2% = (2.15 − 1.69)/1.69 × 100 with respect to the performance of the healthcare monitoring system.

Among all the evaluated algorithms, including the GPR (RMSE 1.95 brpm), the GPR-IPSG had the lowest RMSE value (1.41 brpm). This difference shows that the RMSE was reduced by 0.54 brpm. Hence, a reduction of 0.54 brpm means that the performance improvement of the RR is simply 38.3% = (1.95 − 1.41)/1.41 × 100 based on the BIDMC dataset with respect to the performance of the healthcare monitoring system.

For RMSE, which uses the RRSYNTH dataset, as shown in Table 4, the SVM-IPSG model obtained a lower RMSE value (5.79 brpm) than the SVM model (8.45 brpm). The difference between these two models was 2.66 brpm, which indicated a 45.9% = (8.45 − 5.79)/5.79 × 100 performance improvement regarding the healthcare monitoring system. The NN-IPSG model obtained a lower RMSE value (5.10 brpm) than the NN model (5.83 brpm). The difference between these two models is 0.73 brpm, which denotes a 14.3% = (5.83 − 5.10)/5.10 × 100 performance improvement in the healthcare monitoring system. The GBA-IPSG model outperformed the conventional GBA model, with RMSE values of 6.84 brpm and 8.21 brpm, respectively. The difference between these two models is 1.37 brpm, which indicates a 20.0% = (8.21 − 6.84)/6.84 × 100 performance improvement in the healthcare monitoring system.

The RMSE value of the LSTM-IPSG model (6.15 brpm) was lower than that of the LSTM model (7.89 brpm). The difference between these two models is 1.74 brpm, which denotes a 28.3% = (7.89 − 6.15)/6.15 × 100 performance improvement in the healthcare monitoring system. The RMSE value of the RF-IPSG algorithm (5.36 brpm) was lower than that of the RF model (5.80 brpm), which shows that the RMSE was reduced by 0.44 brpm. Therefore, a reduction of 0.46 brpm means that the performance improvement of the RR is simply 8.2% = (5.80 − 5.36)/5.36 × 100 with respect to the performance of the healthcare monitoring system. As shown in Table 4, the GPR-IPSG model obtained the lowest RMSE value (3.55 brpm), including the GPR model (4.13 brpm). The difference between these two models denotes 0.58 brpm, which indicates a 16.3% = (4.13 − 3.55)/3.55 × 100 performance improvement in the healthcare monitoring system. We show the experimental results using five kernel models based on the GPR algorithm as shown in Table 5. Based on the results in Table 5, the GPR model denotes robust and excellent performance in various kernel environments. Moreover, we obtained 120 samples from (7≤), 150 samples from (8 to 9), 170 samples from (10 to 11), 160 samples from (22 to 23), 130 samples from (24 to 25), and acquired 100 samples from (25≥) brpm samples from 636 original samples, used them as input data for the proposed IPSG algorithm, and acquired 836 artificial samples. Thus, we acquired 1472 samples with 257 feature dimensions. As with the synthetic data results above, in the third experimental scenario, we confirm how the results of the IPSG model and the proposed method were expressed when the ratio of artificial data was changed, as shown in Table 6.

In Figure 7a, the box plot illustrates the distribution and skewness of the MAE results obtained from six ML algorithms. When comparing the MAE values for the SVM, NN, GBA, RF, LSTM, and GPR using the BIDMC dataset, the GPR algorithm yields the lowest MAE, while the LSTM shows the highest MAE. Additionally, the MAEs associated with the LSTM exhibit the most considerable difference between the lower and upper quartiles, indicating that the spread of the middle half of the MAE results is the largest. Figure 7b presents the MAE values for the IPSG model and six combined ML algorithms. In this illustration, the GPR-IPSG model exhibits the lowest MAE values, while the combination of the LSTM-IPSG model shows the highest MAE values. As shown in Figure 7c, the box plot represents the MAE results acquired using six ML algorithms based on the RRSYNTH dataset. The GPR algorithm shows the lowest MAE, while the LSTM and NN show the highest MAEs. The RF algorithm also has a low MAE value. The box plot in Figure 7d denotes the MAE results obtained by combining six ML algorithms and the IPSG model using the RRSYNTH dataset. The GPR-IPSG model has a lower MAE value than the SVM-IPSG, NN-IPSG, GBA-IPSG, RF-IPSG, and LSTM-IPSG models. Additionally, we include the results of subgroups MAE for normal (12–20 brpm), dyspnea (≥20 brpm), and hypopnea (<8 brpm), as shown in Table 7. These results imply that applying the proposed IPSG method in the hypopnea (<8 brpm) subgroup improved performance, as shown in Table 7.

We utilize the analysis of variance (ANOVA) test [46] to evaluate and compare the performance of the proposed IPSG with SVM, NN, GBA, LSTM, RF, and GPR algorithms as shown in Table 8 and Table 9. ANOVA is a statistical method used in all situations requiring comparing two or more population means. That is, the hypothesis of interest in ANOVA is given $H_{0}:\mu_{1}=\mu_{2}\ldots =\mu_{j}$ and $H_{1}:\mu_{1}\neq \mu_{2}\ldots \neq \mu_{j}$. The null hypothesis in ANOVA is that there is no difference in meaning. The alternative hypothesis is that the means are not all equal. Therefore, a multi-comparison was utilized to determine the group averages’ results, which differed from the others. One-way ANOVA is a simple and illustrative example of a linear model, given $e_{ij}=\alpha_{j}+\epsilon_{ij}$. Here, it was assumed that $e_{ij}$ was the experimental result (MAEs) of the proposed IPSG with SVM, NN, GBA, LSTM, RF, and GPR algorithms, where i=30 was the number of measurements, and j=6 denoted the number of groups.

The coefficient of determination ($R^{2}$) is a statistical measure that ranges from 0 to 1. Values closer to 1 indicate a stronger relationship between predictor and response variables. Table 10 shows that the (SVM-IPSG:0.95) has a stronger correlation with the response compared to the (SVM:0.66) as presented in Figure 8a,b. On the other hand, the $R^{2}$ of the (NN:0.53) indicates a correlation with the response variable, which is not stronger when compared to the (NN-IPSG:0.90) as shown in Figure 8c,d. Similarly, the $R^{2}$ of the (GBA-IPSG:0.96) indicates a higher correlation with the response variable than that of the (GBA:0.57) as denoted in Figure 8e,f.

The $R^{2}$ of the (LSTM-IPSG:0.91) has high correlations with the response variables compared to the (LSTM:0.56) as shown in Figure 9a,b. The $R^{2}$ of the (RF-IPSG:0.96) indicates a stronger correlation with the response variable when compared to the (RF:0.71) as shown in Figure 9c,d. The $R^{2}$ of the (GPR-IPSG:0.96) model is the highest of all the models, indicating the strongest correlation with the response variable than that of the (GPR:0.77) model based on the BIDMC dataset, as shown in Figure 9e,f. Figure 9b shows the proposed IPSG artificial data result as a horizontal pattern. This is the artificial label for IPSG estimation.

Based on the RRSYNTH dataset, the (SVM-IPSG:0.78) obtained a stronger correlation with the response compared to the (SVM:0.39) model. The $R^{2}$ of the (NN-IPSG:0.83) denoted a stronger correlation with the response variable when compared to the (NN:0.76). The $R^{2}$ of the (GBA-IPSG:0.67) had a higher correlation with the response variable than that of the (GBA:0.45). The $R^{2}$ of the (LSTM-IPSG:0.74) indicates a high correlation with the response variables compared to the (LSTM-IPSG:0.51) as presented in Table 10. The $R^{2}$ of the (RF-IPSG:0.81) indicates a slightly high correlation with the response variable when compared to the (RF:0.77) as shown in Table 10. The $R^{2}$ of the (GPR-IPSG: 0.92) is the highest of all the models, indicating a higher correlation with the response variable than that of the (GPR:0.89) using the RRSYNTH dataset, as shown in Table 10. Table 11 presents the CIs to indicate the uncertainties associated with the GPR-IPSG and GPR models for estimating RRs using the BIDMC and RRSYNTH dataset.

## 6. Discussion

In this study, we proposed a novel method, imbalanced power spectral generation (IPSG), using a nonparametric bootstrap. This method combines six ML algorithms, including traditional ML and LSTM, to estimate RRs and CIs from PPG signals.

The computational complexity of the GPR-IPSG model was similar to that of the traditional GPR model. Table 2 shows that the LSTM model consumed more computational resources than other ML models. Computational efficiency is an important factor in determining the practical applicability of a method, especially when deployed in real-time systems where speed and resource consumption are important. The key aspects of computational cost include time complexity, space complexity, and overall demand for processing power. However, the limitation of this study is that it focuses on the consumption of computer resources, how fast the six types of algorithms learn data when inputting experimental data, and how fast they estimate the target value (response variable) when inputting test data. According to the evaluation results, the LSTM-IPSG model consumed the most resources at 37.6 s during model training, and GPR-IPSG consumed 1.92 s. The GBA model required 1.22 s for testing, and GPR-IPSG consumed 0.065 s. In addition, the data for training is 1402 continuous data samples and 350 simple samples for testing, which is expected to be sufficient for health monitoring in wearable smart devices.

Among the six evaluated models, the GPR-IPSG model performs best with lower MAE than the GPR algorithm. Except for the SVM-IPSG and LSTM-IPSG combined models, the remaining IPSG combined models show significantly lower MAE results. In addition, the RF-IPSG model also showed low MAE and stable SD. Overall, the proposed IPSG model obtained excellent results based on the BIDMC dataset, as shown in Table 3. The six types of combined models based on the RRSYNTH dataset showed higher MAE results than the combined models using the BIDMC dataset experiments. The artificial data deformation and artifacts in the RRSYNTH dataset cause these results. Many samples in the RRSYNTH dataset were observed to have RRs greater than 50 brpm. These results suggest that the proposed IPSG model is optimized to balance data fit and smoothness accurately. In addition, the proposed IPSG algorithm avoids overfitting, and the generated artificial power spectral curves are fused with the hypopnea imbalance data, which means that the data imbalance is resolved. As a result, the proposed method helps us to accurately estimate RRs as shown in Figure 7b,d.

The RMSE results also showed that the proposed IPSG model performed best when combined with the GPR learning algorithm. The RMSE results of SVM, GBA, NN, and RF models improved when combined with IPSG based on the BIDMC dataset, as shown in Table 4. However, the LSTM model showed insignificant improvement compared to the combined LSTM-IPSG model. According to the RMSE, results outperformed based on the RRSYNTH dataset. Although the RMSE result of the SVM algorithm (8.45 brpm) was worse than five algorithms such as NN, GBA, LSTM, RF, and GPR, when the SVM learning model was combined with the IPSG algorithm, the RMSE result (5.79 brpm) outperformed the GBA-IPSG (6.84 brpm) and LSTM-IPSG (6.15 brpm) algorithms because the number of imbalanced data significantly increased. Therefore, when the proposed IPSG model is combined with the conventional ML models, we confirm that it shows a lower RMSE than the conventional ML model without the IPSG model on the RRSYNTH dataset.

In addition, we presented the results of MAE using five kernel models based on GPR and GPR-IPSG in the second experimental scenario, as shown in Table 5. The table shows that GPR and the proposed IPSG models are very stable and robust and show excellent results despite the various kernel changes, implying the high reliability of the GPR and IPSG algorithms. In the third experimental scenario, the results of the IPSG and the ML models, relating to the changed ratio of artificial data, can be seen in Table 6. The NN-IPSG method showed the best results in MAE, RMSE, and $R^{2}$. The remaining algorithms also showed excellent and stable results. The NN method showed many changes in the results due to adjusting the ratio of artificial synthetic data, so the NN algorithm may be considered sensitive.

Regarding the MAE results denoted in Table 7, we found that in normal breathing situations (12–20 brpm), the results of MAE were almost similar or that the performance of ML decreased when IPSG was applied. In addition, when IPSG was applied in dyspnea situations, the performance of ML was improved. GBA-IPSG (1.19 brpm) showed better results than GBA (1.94 brpm). NN-IPSG and GPR-ISPG also showed excellent results. In particular, applying the IPSG method to all ML algorithms in hypopnea (<8 brpm) situations showed much better results than when only the ML algorithm was used. These results indicate surprising changes when clinical data are sparse, such as hypopnea (<8 brpm).

Based on the results of the ANOVA test on the left, as shown in Table 8, the performance of six ML algorithms was analyzed. Table 8 presents the between-groups variation (Group) and within-groups variation (Errors), where SS is the sum of squares, and df is the degrees of freedom. The total degree of freedom is the total number of measurements (MAEs) minus one, which denotes 179 (=180 − 1). The between-groups degrees of freedom are the number of groups minus one, which denotes 5 (=6 − 1). The MS denotes the mean squared error (2.59), which denotes SS (12.96)/df (5). The F-statistic denotes the ratio of the mean squared errors (MS/Error). The *p*-value, 1.11 × $10^{-44}$, denotes the probability that the test statistic can obtain a value greater than the value of the calculated test statistic, i.e., P (F > 149.26). The small *p*-value, 1.11 × $10^{-44}$ < (α=0.05), denotes that differences between group means are significant. As shown in the right part of Table 8, the *p*-value, 5.1 × $10^{-46}$, is less than the significant value (0.05). We confirm the results of the ANOVA test on the left, as shown in Table 9; the very small *p*-value, 5.33 × $10^{-61}$ < (α=0.05), indicates that the differences between group means are significant. The results of the ANOVA test on the right, as shown in Table 9, the *p*-value, 9.04 × $10^{-67}$ < (α=0.05), indicates that differences between group means are significant.

Additionally, the R2 values of the SVM-IPSG and NN-IPSG algorithms (0.95, 0.90) in Figure 8b,d showed a stronger relationship with the response than the $R^{2}$ values of SVM and NN without the IPSG algorithm (0.66, 0.53) as shown in Figure 8a,c; (see Table 10). The $R^{2}$ (0.90) of the NN-IPSG algorithm denoted higher relationships with the response variables than the $R^{2}$ (0.53) of the NN algorithm, as shown in Table 10 and Figure 8e,f. The $R^{2}$ (0.96) of the GBA-IPSG algorithm also denoted a stronger correlation with the response variables than the $R^{2}$ (0.57) of the GBA algorithm when using the BIDMC dataset with the first experimental scenario, as shown in Figure 9e,f. These results denote that applying the proposed ISPG method in the hypopnea (<8 brpm) subgroup improved performance, as shown in Table 7. The $R^{2}$ (0.91) of the LSTM-IPSG algorithm presented higher relationships with the response variables than the $R^{2}$ (0.56) of the LSTM algorithms, as shown in Table 10 and Figure 9a,b.

In Figure 9c,d, the $R^{2}$ (0.96) of the RF-IPSG algorithm also showed higher correlations with the response variables than the $R^{2}$ (0.71) of the conventional RF algorithms when using the BIDMC dataset (see Table 10). The $R^{2}$ (0.96) of the GPR-IPSG algorithm also presented higher correlations with the response variables than the $R^{2}$ (0.77) of conventional GPR algorithms when using the BIDMC dataset as denoted in Figure 9e,f; (see Table 10). In the RRSYNTH dataset experiments, we confirmed that combining the IPSG algorithm with the conventional learning model improved performance (see Table 10). Therefore, it indicates that the proposed IPSG algorithm can improve the performance of six learning algorithms by solving the imbalanced data problem.

We utilized GPR and GPR-IPSG models to estimate the respiration rate (RR) uncertainty. The respiration rate (RR) reflects the variability of the RR estimate. Table 11 shows that GPR and GPR-IPSG models can independently generate CI without combining other algorithms. In contrast, conventional machine learning models usually need to integrate other algorithms to estimate uncertainty and derive CI. The CI results of the GPR and GPR-IPSG models are shown in Figure 10. Although these models effectively estimate the uncertainty of RRs, they have difficulty accurately assessing low and high breaths per minute (brpm) due to the imbalanced data in the RR and CI estimates. The GPR-IPSG method is effective for estimating both low and high brpm respiration and for assessing uncertainty by measuring the CI based on the BIDMC dataset, as illustrated in Figure 10b. This proposed method addresses the issue of imbalanced data and enhances overall performance. Typically, devices that measure RRs provide only single-point estimates without a CI, making it challenging to differentiate between statistical and physiological variations. As a result, the IPSG can be utilized in home monitoring systems for elderly patients. Frequent and widespread CIs in elderly individuals can trigger alarms or notify nurses or primary care physicians in home monitoring settings [47]. Thus, predicting the CIs for RR measurements can significantly improve reliability.

The GPR and GPR-IPSG models differed minimally from the reference RR values. The agreement limits are indicated by the red horizontal lines in Figure 11a,b are mean error (ME) ±2×SD. Figure 11a shows the Bland–Altman plot of reference RR, with the ME of 0.13 mmHg and the SD ± 1.96 brpm. Figure 11b presents the Bland–Altman plot of the reference RR, with the ME of −0.03 brpm and the SD ± 1.40 brpm. The SDs of RF and RF-IPSG are clustered tightly within a narrow range compared to the baseline RR. The reliability of the RF and RF-IPSG algorithms is highlighted in Figure 11c,d, which compares the SDs (±2.46 brpm and ±1.82 brpm) of the MEs (0.19 brpm and 0.08 brpm) of RF and RF-IPSG with the baseline RR. The SDs of RF and RF-IPSG show consistent and reliable performance compared to the baseline RR, which ensures their performance. The Brand-Artman plots of GBA and GBA-IPSG in Figure 11e,f also show that they are located within the small MEs (−0.23 brpm and −0.18 brpm) and narrow SDs (±2.76 brpm and ±1.84 brpm) compared to the baseline RR. These results also imply reliable results and confirm that GPR-IPSG, RF-IPSG, and GBA IPSG agree better with the reference RR than GPR, RF, and GBA. This is because the ME of the estimated RR using the proposed IPSG methodology is close to 0, and most blue and black points are within the upper and lower bounds.

In the second experiment with the RRSYNTH dataset, as shown in Table 11, the CIs were extensive, which indicates that the dataset is unsuitable for estimating RR fluctuations. These results suggest that the estimation errors, measured by the MAE and RMSE, increase due to signal distortion and artifacts in the RRSYNTH dataset. Consequently, the CIs also expand. Moving forward, we will continue to explore methods to reduce RR estimation errors through various data experiments and aim to establish CIs that can be applied for reliable home-based monitoring. We tested and evaluated the performance of the proposed IPSG algorithm using two datasets. However, the clinical BIDMC dataset has limitations, including a relatively small number of participants and a limited number of records. Additionally, while the RRSYNTH dataset contains sufficient recorded samples, the estimated RR was greater than 40 brpm due to significant artificial data artifacts.

The primary challenge in developing a home-based monitoring device lies in the variability of PPG (photoplethysmography) signals among patients. These signals can differ based on factors such as the measurement location, skin texture, and motion artifacts. Additionally, variations occur due to the cardiovascular characteristics of elderly patients. In this paper, we introduced a bootstrap-based method for generating power spectral data, which is independent of these factors. Our findings demonstrated that this new approach performs equally well for all patients. The GPR-IPSG model achieved a coefficient of determination of 0.96, which indicates the strongest correlation with the response variable among all models tested. This result signifies an improvement in performance compared to existing ML models that do not utilize the IPSG method, as illustrated in Table 3, Table 4, Table 5, Table 6 and Table 7.

Future research will focus on the issues mentioned above to improve robustness against noise, such as motion artifacts, and to reduce the complexity of the device using the PPG signal. Another area we aim to enhance is the reliability of critical healthcare devices. While we have validated our proposed method using the publicly available BIDMC and RRSYNTH datasets, we plan to validate our IPSG method further using the heterogeneous standard database MIMIC-III, which contains 12,000 records representing diverse clinical cases. This comprehensive database validation will ensure our methodology’s reliability for clinical applications. Ultimately, this will improve life monitoring systems’ reliability in mobile and home-based healthcare devices. Although the BIDMC data do not contain respiration rates above 25 brpm, which is a limitation of the BIDMC dataset, we completed the experiment using the RRSYNTH dataset. Although the RRSYNTH data also completed the experiment by generating artificial data up to about 25 brpm, we plan to experiment by generating artificial data above 30 brpm in future studies.

The CI represents the RR’s uncertainty for reliability improvement. The CI obtained from the RRSYNTH dataset is calculated based on the distribution of RR estimates, resulting in a relatively wide CI. This wider span occurs because the error in the RR estimate increases compared to data derived from BIDMC. To address the wide CI in the RRSYNTH dataset, which contains various artificial distortions, we will focus on reducing the robust RR estimate error. A wider CI increases the likelihood that RR estimates will be included; however, it also lowers the reliability of the RR monitoring system. Conversely, if the CI is too narrow, a slight change in the RR estimate could easily fall outside the CI, limiting its effectiveness as an RR monitoring tool. Therefore, we will strive to establish an appropriate range for the CI in future studies to overcome these limitations.

Moreover, while we utilized single PPG signals to simplify the hardware needed for continuous biosignal monitoring, we must also consider power efficiency for mobile applications. Currently, machine learning and deep learning approaches are computationally intensive. To address this, we plan to develop a simpler machine-learning model to enhance algorithm efficiency and reduce device complexity.

## 7. Conclusions

In this study, we propose a new method for improving the accuracy of RR estimation by applying the proposed IPSG model to six conventional learning algorithms, using PPG signals and combining the CIs that can determine RR fluctuations with the GPR model. The proposed method segments the input PPG signal and extracts continuous features, using the autocorrelation function-based power spectral feature extraction technique. The imbalanced distribution features of the PPG signal, which were the focus of this study, were supplemented by generating artificial features using the nonparametric bootstrap technique. We trained the GPR-IPSG model by combining the original feature curve and the artificially generated feature curve to overcome the overfitting and data sample imbalance problems. This approach achieved high stability and accuracy by randomly mixing the original and artificial feature curves. The proposed GPR-IPSG algorithm measures a CI, which represents the level of uncertainty that may occur due to the physiological variability of the PPG signal. Therefore, the proposed GPR algorithm is a reliable model for estimating RRs and CIs accurately. Future studies will be conducted to expand the dataset, use cross-validation with a larger cohort, and apply the proposed method to heart rate and blood pressure variability. In addition, we will validate the proposed technique using another public dataset (MIMIC-III). Therefore, the proposed GPR-IPSG model can be used to improve the performance of clinical home-based monitoring systems and design a reliable framework. These simple wearable devices can continuously estimate RRs and CIs and are convenient to wear and carry for long periods.

## Figures and Tables

**Figure 1 sensors-25-01437-f001:**
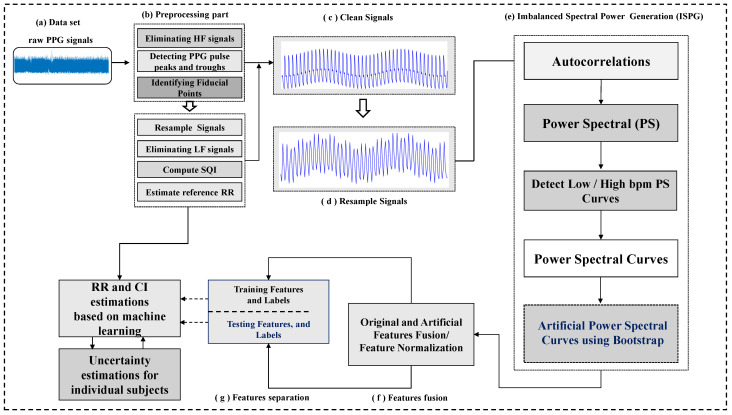
The proposed methodology with ML algorithms for RRs and uncertainty estimations, where the following holds: (**a**) denotes a raw PPG signal; (**b**) is a preprocessing part; (**c**) denotes a clean signal; (**d**) resample signal; (**e**) bootstrap-based imbalanced feature generation process; (**f**) feature fusion; (**g**) feature separation.

**Figure 2 sensors-25-01437-f002:**
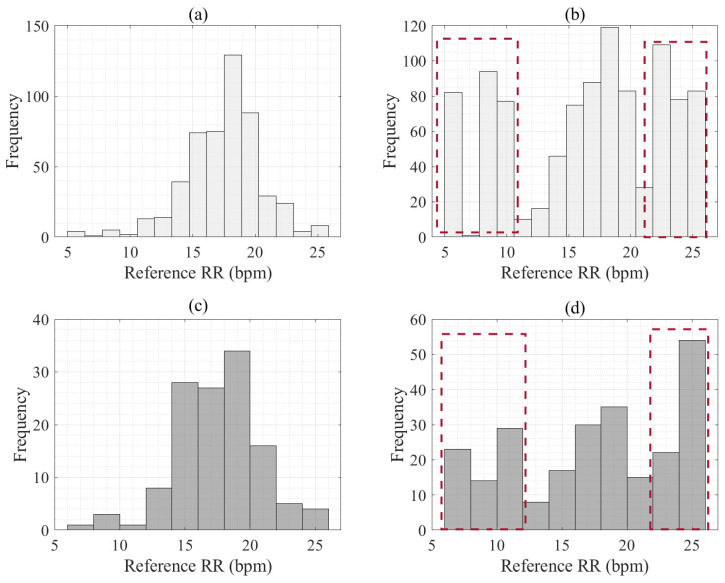
The distribution of reference RR values. Panel (**a**) shows the distribution of reference RR for training. Panel (**b**) is the extended distribution of reference RR using IPSG for training. The red dotted line is the augmented artificial IPSG-related reference RR for training. Panel (**c**) shows the distribution of reference RR for testing. Panel (**d**) is the extended distribution of reference RR using IPSG for testing. The red dotted line shows the augmented artificial IPSG-related reference RR for testing.

**Figure 3 sensors-25-01437-f003:**
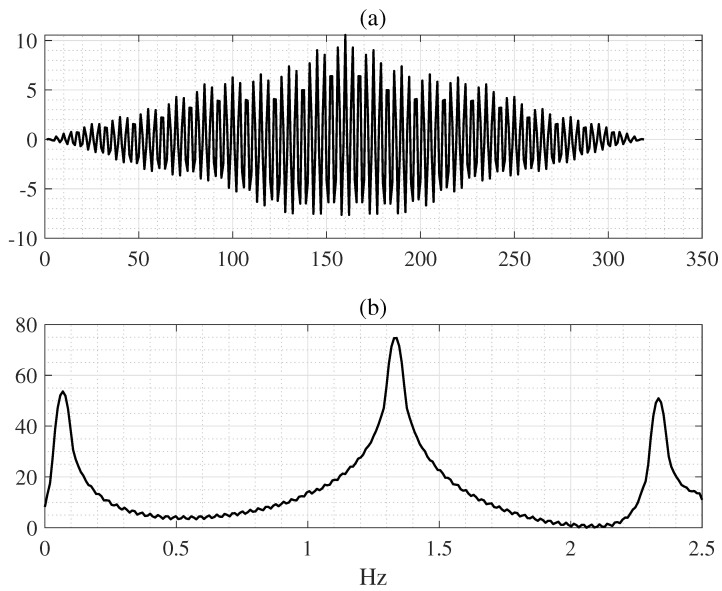
(**a**) The upper panel is autocorrelation. (**b**) The bottom panel denotes the power spectral (PS) curve from the RRSYNTH dataset.

**Figure 4 sensors-25-01437-f004:**
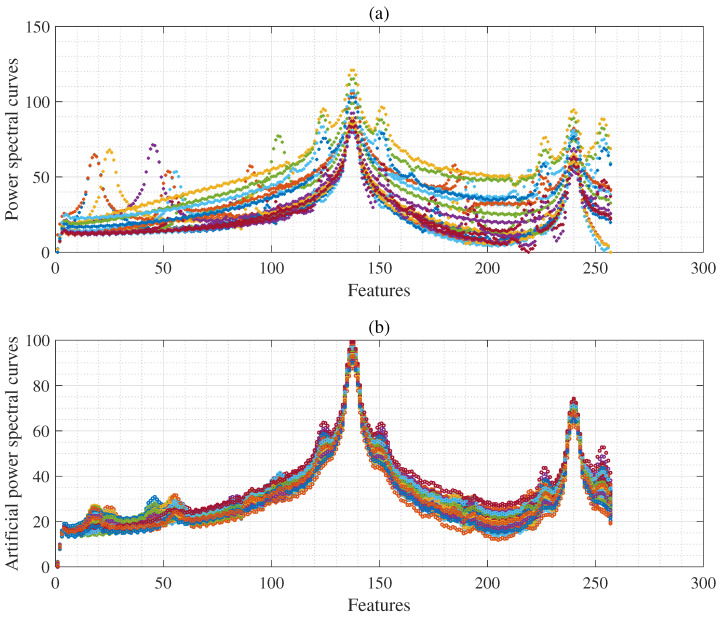
(**a**) In the upper panel are the original power spectral (PS) curves from the autocorrelation (example: 24 brpm). (**b**) The bottom panel denotes the artificial power spectral (PS) curves using bootstrap from the power spectral (PS) curves (**a**).

**Figure 5 sensors-25-01437-f005:**
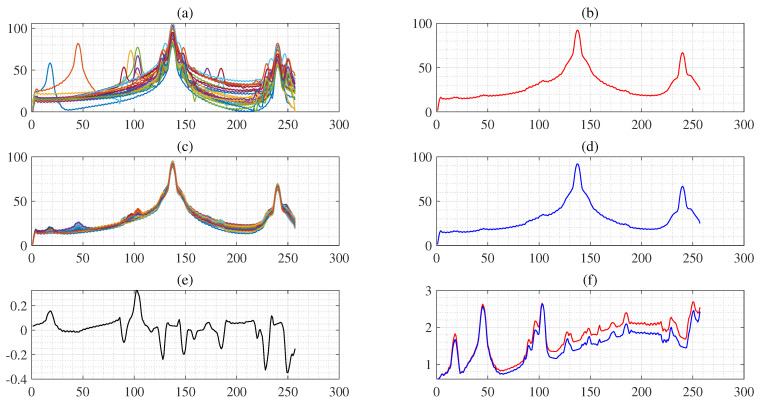
Panel (**a**) denotes each of the PS feature curves, panel (**b**) is the mean of the PS feature curves, panel (**c**) denotes each of the artificial PS feature curves using bootstrap, panel (**d**) is the mean of the artificial PS feature curves using bootstrap, panel (**e**) denotes the bias between the mean of the PS feature curves and the mean of the artificial PS feature curves using bootstrap, the red line of panel (**f**) is the standard error (SE) of the PS feature curves, and the blue line of the panel (**f**) is SE of the artificial PS feature curves using bootstrap.

**Figure 6 sensors-25-01437-f006:**
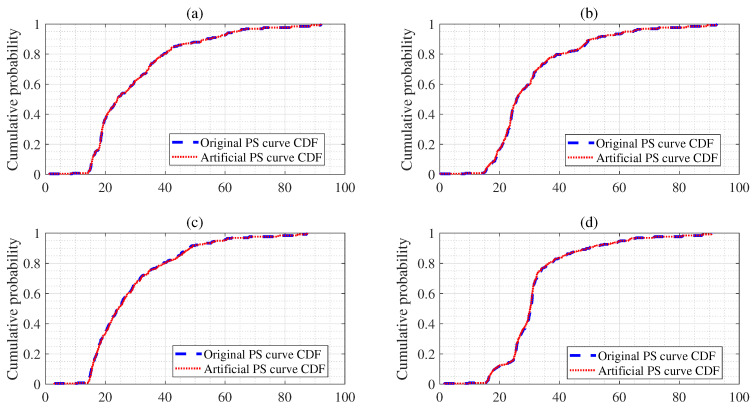
Panel (**a**) is mean of feature vectors (≤brpm 7), panel (**b**) denote mean of feature vectors (=brpm 10), panel (**c**) is mean of feature vectors (=brpm 24), and panel (**d**) denote mean of feature vectors (≥brpm 25).

**Figure 7 sensors-25-01437-f007:**
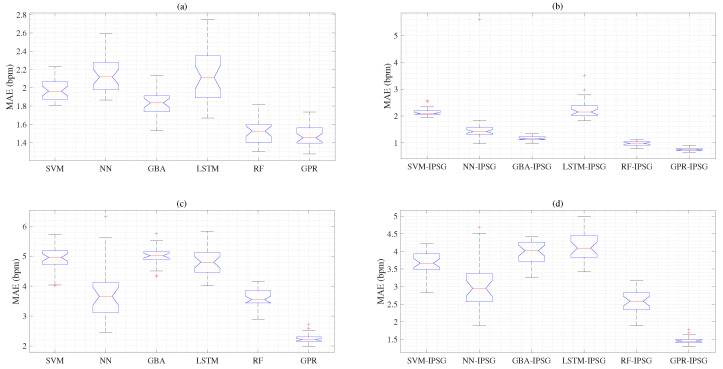
Panel (**a**) represents the MAE and SD compared to the reference RR obtained from the SVM, NN, GBA, RF, LSTM, and GPR models based on the BIDMC dataset. Panel (**b**) shows the MAE and SD compared to the reference RR obtained from the SVM-IPSG, NN-IPSG, GBA-IPSG, LSTM-IPSG, RF-IPSG, and GPR-IPSG models, using the BIDMC dataset. Panel (**c**) denotes the MAE and SD compared to the reference RR obtained from the SVM, NN, GBA, LSTM, RF, and GPR algorithms, using the RRSYNTH dataset. Panel (**d**) denotes the MAE and SD compared to the reference RR acquired from the SVM-IPSG, NN-IPSG, GBA-IPSG, LSTM-IPSG, RF-IPSG, and GPR-IPSG models, using the RRSYNTH dataset.

**Figure 8 sensors-25-01437-f008:**
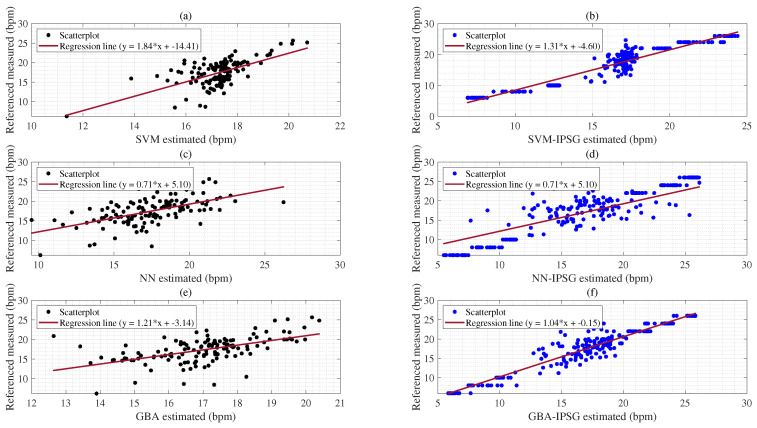
Top panel (**a**) represents a $R^{2}$ results compared with the reference RR value acquired from the SVM; Panel (**b**) denotes a $R^{2}$ results compared with the reference RR value obtained from the SVM-IPSG; Panel (**c**) shows a $R^{2}$ results compared to the reference RR acquired from the NN; Panel (**d**) denotes a $R^{2}$ results compared to the reference RR acquired from the NN-IPSG; Panel (**e**) denotes a $R^{2}$ results compared to the reference RR acquired from the GBA; Panel (**f**) denotes a $R^{2}$ results compared to the reference RR acquired from the GBA-IPSG based on the BIDMC dataset.

**Figure 9 sensors-25-01437-f009:**
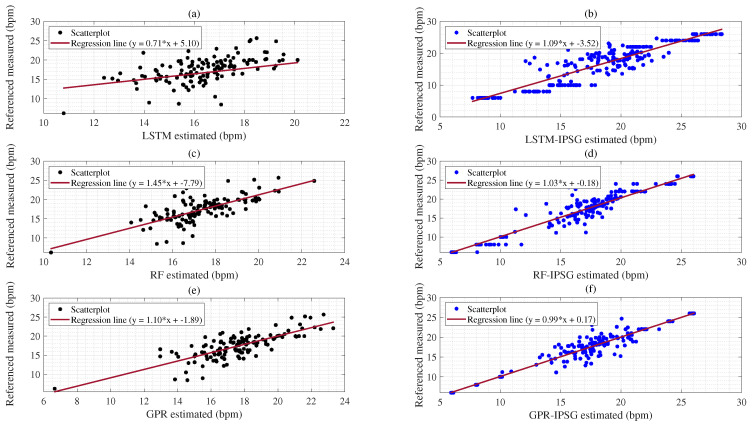
Top panel (**a**) represents $R^{2}$ results compared with the reference RR value acquired from the LSTM; Panel (**b**) denotes $R^{2}$ results compared with the reference RR value obtained from the LSTM-IPSG; Panel (**c**) shows $R^{2}$ results compared to the reference RR acquired from the RF; Panel (**d**) denotes $R^{2}$ results compared to the reference RR acquired from the RF-IPSG; Panel (**e**) denotes a $R^{2}$ results compared to the reference RR acquired from the GPR; Panel (**f**) denotes a $R^{2}$ results compared to the reference RR acquired from the GPR-IPSG based on the BIDMC dataset.

**Figure 10 sensors-25-01437-f010:**
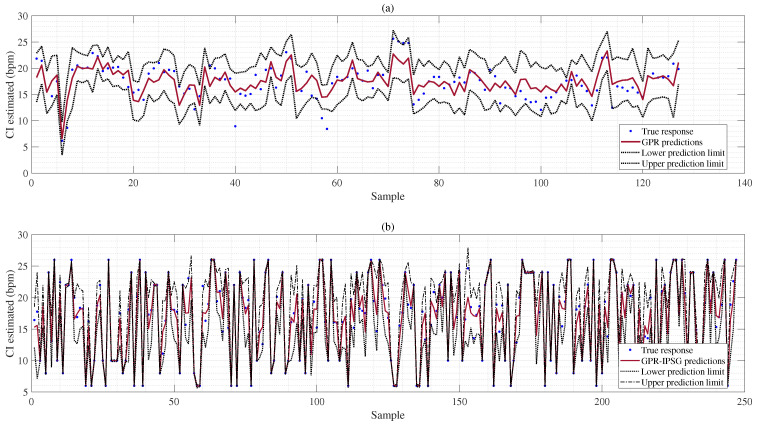
Panel (**a**) denotes the for RR estimation; panel (**b**) is based on the BIDMC dataset.

**Figure 11 sensors-25-01437-f011:**
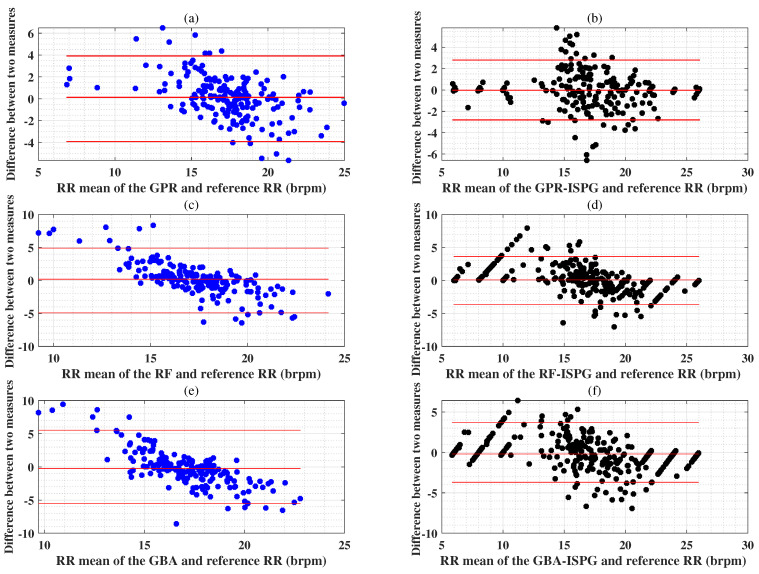
The top panel (**a**,**b**) display Bland–Altman plots illustrating the difference between the RR mean and the reference value for GPR and GPR-IPSG regarding RR estimations; Panel (**c**,**d**) display Bland–Altman plots illustrating the difference between the RR mean and the reference value for RF and RF-IPSG regarding RR estimations; Panel (**e**,**f**) display Bland–Altman plots illustrating the difference between the RR mean and the reference value for GBA and GBA-IPSG regarding RR estimations utilizing the BIDMC dataset in the second experimental scenario.

**Table 1 sensors-25-01437-t001:** Summary parameters for conventional ML and the proposed IPSG methods using the public RRSYNTH dataset [31]. These results were obtained using the first experimental scenario.

Methods	SVM [19]	NN [44]	GBA [11]	LSTM [18]	RF [41]	GPR [42]
Combined	IPSG	IPSG	IPSG	IPSG	IPSG	IPSG
Dataset: RRSYNTH						
Number of Training Sample	1402	1402	1402	1402	1402	1402
Number of Testing Sample	350	350	350	350	350	350
Number of Feature	160	160	160	160	160	160
Output Dimension	1	1	1	1	1	1
Learner			tree		tree	
Learning Cycle					30	
Shrinkage Factor			0.1			
Subsampling Factor			0.1			
Max Tree Depth			uint32(6)		8	
Rand Seed			uint32(rand()*1000)			
Number of Iteration			1000			
KFold	5			5	5	5
Layer Size		10				
Activations		Relu		Relu		
Lambda		0				
Standardize	False	False		False		
Optimizer						Quasi Newton
Box Constraint						
Epsilon	3.0					
Kernel Scale	auto					
Solver	ISDA			adam		
Kernel Function	Gaussian					squareexponential
Basic Function						none
Fit Method						exact
Predict Method						exact
Active Set Method						random
Number of HiddenUnits				500		
Layer				Sequence Input Layer		
				Lstm Layer		
				Fully Connected Layer (50)		
				Dropout Layer (0.5)		
				Fully Connected Layer(1)		
				Regression Layer		
Max Epochs				200		
Initial Learn Rate				0.01		
Mini Batch Size				1		
Gradient Threshold				1		
Drop Factor				0.5		
Shuffle				Never		
Loss			Squaredloss	MSE		
Sequence Length				Longest		
Metrics				RMSE		
Verbose	0	0		0		0

**Table 2 sensors-25-01437-t002:** Comparing training and testing times between the conventional and proposed methodologies using Intel Core i5-9400 CPU 4.1 GHz, OS 64 bit, RAM 16.0 GB, and Matlab 2023 (The MathWorks Inc., Natick, MA, USA) system specifications based on the RRSYNTH dataset, where IPSG denotes the proposed IPSG model. These results were obtained using the first experimental scenario.

Methods	SVM	SVM	NN	NN	GBA	GBA	LSTM	LSTM	RF	RF	GPR	GPR
Combined		IPSG		IPSG		IPSG		IPSG		IPSG		IPSG
Training time (s)	0.08	0.28	2.20	2.35	2.24	3.10	21.58	37.60	1.34	2.35	1.88	1.92
Testing time (s)	0.01	0.05	0.02	0.02	1.22	0.92	0.09	0.60	0.03	0.03	0.07	0.06

**Table 3 sensors-25-01437-t003:** Using multiple PPG-based ML algorithms, we computed them as the differences from the reference RR values in order to express it as the MAE and SD results, where 1st denotes the first experimental scenario, and 2nd is the second experimental scenario.

Dataset	Error (brpm)	SVM	SVM	NN	NN	GBA	GBA	LSTM	LSTM	RF	RF	GPR	GPR
	Combined		IPSG		IPSG		IPSG		IPSG		IPSG		IPSG
BIDMC	MAE (1st)	1.98	2.15	2.15	1.57	1.84	1.17	2.15	2.22	1.53	0.98	1.47	0.79
	SD (1st)	0.12	0.14	0.18	0.78	0.14	0.09	0.35	0.41	0.13	0.09	0.12	0.06
	MAE (2nd)	2.05	1.46	2.29	1.41	1.88	1.60	2.37	1.57	1.58	0.95	1.50	1.21
	SD (2nd)	0.09	0.09	0.15	0.28	0.11	0.10	0.48	0.19	0.10	0.06	0.08	0.09
RRSYNTH	MAE (1st)	4.95	3.66	3.78	3.02	5.03	3.98	4.85	4.11	3.60	2.60	2.24	1.47
	SD (1st)	0.40	0.30	0.91	0.73	0.31	0.33	0.49	0.39	0.29	0.32	0.16	0.10
	MAE (2nd)	5.21	4.79	9.96	7.86	5.34	5.03	7.09	6.49	4.02	3.88	3.46	3.09
	SD (2nd)	0.46	0.34	2.07	1.28	0.38	0.26	0.58	0.65	0.32	0.24	0.26	0.21

**Table 4 sensors-25-01437-t004:** Using the PPG-based multiple algorithms are computed as the difference from the reference RR values to express it as the RMSE and SD results.

Dataset	Error (brpm)	SVM	SVM	NN	NN	GBA	GBA	LSTM	LSTM	RF	RF	GPR	GPR
	Combined		IPSG		IPSG		IPSG		IPSG		IPSG		IPSG
BIDMC	RMSE (1st)	2.29	1.88	2.57	2.32	2.50	1.73	2.54	2.47	2.15	1.69	1.95	1.41
	SD (1st)	0.15	0.17	0.28	0.84	0.23	0.13	0.24	0.21	0.22	0.15	0.18	0.14
	RMSE (2nd)	2.38	2.30	2.66	2.76	2.57	2.19	2.64	2.73	2.22	2.11	2.00	1.78
	SD (2nd)	0.17	0.19	0.19	0.50	0.22	0.13	0.19	0.15	0.17	0.15	0.12	0.12
RRSYNTH	RMSE (1st)	8.45	5.79	5.83	5.10	8.21	6.84	7.89	6.15	5.80	5.36	4.13	3.55
	SD (1st)	0.70	0.91	1.08	1.05	0.60	0.84	0.87	0.80	0.63	0.94	0.54	0.77
	RMSE (2nd)	8.54	7.20	9.16	8.78	8.55	7.64	9.19	8.89	6.39	6.15	6.31	5.80
	SD (2nd)	0.74	0.56	0.72	0.62	0.66	0.52	0.75	0.66	0.71	0.43	0.61	0.50

**Table 5 sensors-25-01437-t005:** We present the results of experiments using various kernel models for the GPR algorithm.

Dataset	Error (brpm)	GPR	GPR	GPR	GPR	GPR	GPR	GPR	GPR	GPR	GPR
	Combined		IPSG		IPSG		IPSG		IPSG		IPSG
	Kernel	Expon.	Expon.	Squareexp.	Squareexp.	Matern32	Matern32	Matern52	Matern52	Rational.	Rational.
BIDMC	MAE (2nd)	1.49	1.20	1.49	1.24	1.47	1.27	1.47	1.28	1.46	1.24
	SD (2nd)	0.06	0.08	0.06	0.06	0.10	0.06	0.06	0.06	0.06	0.06
	RMSE (2nd)	1.97	1.70	2.00	1.81	1.97	1.83	1.98	1.85	1.97	1.81
	SD (2nd)	0.09	0.10	0.09	0.12	0.09	0.12	0.09	0.12	0.09	0.12
RRSYNTH	MAE (2nd)	3.57	3.38	3.48	3.10	3.41	3.10	3.39	3.04	3.41	3.06
	SD (2nd)	0.29	0.27	0.26	0.30	0.27	0.28	0.27	0.28	0.28	0.28
	RMSE (2nd)	6.85	6.08	6.86	6.11	6.62	5.81	6.61	5.80	6.59	5.81
	SD (2nd)	0.62	0.53	0.55	0.57	0.61	0.55	0.59	0.55	0.60	0.55

**Table 6 sensors-25-01437-t006:** The results are denoted by varying the proportion of imbalanced data.

Dataset	Error (brpm)	SVM	SVM	NN	NN	GBA	GBA	LSTM	LSTM	RF	RF	GPR	GPR
	Combined		IPSG		IPSG		IPSG		IPSG		IPSG		
BIDMC	MAE (2nd)	1.91	1.49	1.75	0.67	1.86	1.12	2.24	2.11	1.57	0.90	1.49	0.73
	SD (2nd)	0.27	0.09	0.09	0.05	0.10	0.06	0.40	0.43	0.13	0.06	0.09	0.05
	RMSE (2nd)	2.65	2.26	2.31	1.35	2.52	1.74	2.61	2.30	2.19	1.62	2.00	1.41
	SD (2nd)	0.30	0.11	0.14	0.09	0.15	0.10	0.16	0.20	0.19	0.13	0.12	0.08
	$R^{2}$ (2nd)	0.54	0.93	0.67	0.98	0.58	0.96	0.52	0.93	0.79	0.97	0.77	0.97
	SD (2nd)	0.05	0.01	0.06	0.15	0.04	0.01	0.07	0.01	0.04	0.01	0.03	0.01

**Table 7 sensors-25-01437-t007:** The MAE results show different respiratory subgroups as hypopnea (<8 brpm), normal (12–20 brpm), and dyspnea (≥20 brpm).

Dataset	Error (brpm)	SVM	SVM	NN	NN	GBA	GBA	LSTM	LSTM	RF	RF	GPR	GPR
	Combined		IPSG		IPSG		IPSG		IPSG		IPSG		
BIDMC	MAE (hypopnea)	7.32	0.64	6.01	0.08	8.58	0.43	7.12	1.24	7.90	0.29	2.45	0.09
	SD (hypopnea)	0.93	0.09	1.14	0.03	0.87	0.12	1.65	0.81	0.88	0.21	1.08	0.05
	MAE (normal)	1.12	1.34	1.37	1.51	1.26	1.56	1.75	2.56	1.12	1.34	1.27	1.58
	SD (normal)	0.07	0.68	0.08	0.12	0.07	0.10	0.35	0.33	0.06	0.09	0.07	0.09
	MAE (dyspnea)	1.94	0.99	2.30	0.46	1.94	1.19	3.39	2.45	2.30	0.84	1.94	0.50
	SD (dyspnea)	0.31	0.11	0.30	0.07	0.31	0.15	1.22	1.04	0.30	0.14	0.31	0.08

**Table 8 sensors-25-01437-t008:** The ANOVA test results on the left are obtained from 30 MAE experiments of the SVM, NN, GBA, LSTM, RF, and GPR algorithms. The ANOVA test results on the right are obtained from 30 MAE experiments, using the SVM-IPSG, NN-IPSG, GBA-IPSG, LSTM-IPSG, RF-IPSG, and GPR-IPSG algorithms. Both experimental results are from the BIDMC dataset.

Source	SS	df	MS	F	Prob. > F (*p*-Value)	Source	SS	df	MS	F	Prob. > F (*p*-Value)
Group	12.96	5	2.59	84.44	1.1 × $10^{-44}$	Group	57.93	5	11.59	88.76	5.1 × $10^{-46}$
Error	5.34	174	0.03			Error	22.71	174	0.13		
Total	18.30	179				Total	80.64	179			

**Table 9 sensors-25-01437-t009:** The ANOVA test results on the left are obtained from 30 MAE experiments of the SVM, NN, GBA, LSTM, RF, and GPR algorithms. The ANOVA test results on the right are obtained from 30 MAE experiments of the SVM-IPSG, NN-IPSG, GBA-IPSG, LSTM-IPSG, RF-IPSG, and GPR-IPSG algorithms. Both experimental results are from the RRSYNTH dataset.

Source	SS	df	MS	F	Prob. > F (*p*-Value)	Source	SS	df	MS	F	Prob. > F (*p*-Value)
Group	177.99	5	35.60	149.26	5.33 × $10^{-61}$	Group	150.43	5	30.09	179.75	9.04 × $10^{-67}$
Error	41.5	174	0.24			Error	29.12	174	0.17		
Total	219.49	179				Total	179.55	179			

**Table 10 sensors-25-01437-t010:** Using the multiple PPG-based algorithms, we computed them as the differences from the reference RR values in order to express it as the $R^{2}$ and SD results.

Dataset	Error (brpm)	SVM	SVM	NN	NN	GBA	GBA	LSTM	LSTM	RF	RF	GPR	GPR
	Combined		IPSG		IPSG		IPSG		IPSG		IPSG		IPSG
BIDMC	$R^{2}$ (1st)	0.66	0.95	0.53	0.90	0.57	0.96	0.56	0.91	0.71	0.96	0.77	0.96
	SD (1st)	0.05	0.01	0.12	0.17	0.06	0.01	0.08	0.02	0.09	0.01	0.03	0.01
	$R^{2}$ (2nd)	0.64	0.90	0.52	0.83	0.56	0.91	0.53	0.85	0.79	0.91	0.76	0.94
	SD (2nd)	0.06	0.02	0.06	0.15	0.06	0.01	0.05	0.02	0.04	0.01	0.03	0.01
RRSYNTH	$R^{2}$ (1st)	0.39	0.78	0.76	0.83	0.45	0.67	0.51	0.74	0.77	0.81	0.89	0.92
	SD (1st)	0.08	0.05	0.11	0.08	0.07	0.06	0.10	0.05	0.05	0.05	0.03	0.03
	$R^{2}$ (2nd)	0.36	0.60	0.08	0.21	0.37	0.53	0.06	0.16	0.71	0.73	0.72	0.76
	SD (2nd)	0.09	0.03	0.06	0.08	0.04	0.03	0.05	0.07	0.07	0.02	0.07	0.04

**Table 11 sensors-25-01437-t011:** We compare the CIs obtained, using the GPR and GPR-IPSG, where U denotes the upper limit, and L denotes the lower limit.

Methods	GPR	GPR IPSG	GPR	GPR IPSG	GPR	GPR IPSG	GPR	GPR IPSG
Dataset	BIDMC	BIDMC	RRSYNTH	RRSYNTH	BIDMC	BIDMC	RRSYNTH	RRSYNTH
	(1st)	(1st)	(1st)	(1st)	(2nd)	(2nd)	(2nd)	(2nd)
RR (SD)	17.40 (0.15)	16.83 (0.32)	20.93 (0.44)	19.26 (0.44)	15.06 (0.28)	16.83 (0.32)	20.93 (0.44)	19.26 (0.44)
RR (SD) CI L	13.43 (0.20)	14.71 (0.30)	10.75 (0.39)	11.99 (0.45)	11.82 (0.28)	14.71 (0.30)	10.75 (0.39)	11.99 (0.45)
RR (SD) CI U	21.38 (0.24)	18.95 (0.41)	31.11 (0.64)	26.54 (0.56)	18.29 (0.49)	18.95 (0.41)	31.11 (0.64)	26.54 (0.56)
RR (SD) 95% CI	7.95 (0.32)	4.23 (0.11)	20.37 (0.59)	14.56 (0.71)	6.47 (0.30)	4.23 (0.11)	20.37 (0.59)	14.56 (0.71)

## Data Availability

The original data presented in the study are openly available at http://peterhcharlton.github.io/RRest/bidmcdataset.html and http://peterhcharlton.github.io/RRest/syntheticdataset.html.

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
