# Peer review of "Imbalanced Power Spectral Generation for Respiratory Rate and Uncertainty Estimations Based on Photoplethysmography Signal"

_sensors, 2025, doi:10.3390/s25051437_

Round 1

Reviewer 1 Report

Comments and Suggestions for Authors

Abstract: While the abstract mentions improvements in accuracy and uncertainty management, it lacks a clear conclusion summarizing the impact or implications of the findings.

Recommendations:

-Include a concise conclusion at the end of the abstract that synthesizes the key findings and their significance in clinical enviroment.

Methods: The study appropriately employs relevant datasets (MIMIC-II and RRSYNTH) and methods (e.g., imbalanced feature generation using bootstrapping), aligning with its objective of evaluating a proposed algorithm for respiratory rate estimation. Signal processing steps and feature generation procedures are well-documented, demonstrating a clear intervention by the authors.

Recommendations: 

-Conduct a statistical significance analysis to determine whether the proposed method outperforms existing approaches.

-Include a computational cost analysis and assess the feasibility of real-time implementation.

Data Availability Statement: The provided link to the Data Availability Statement is not accessible.

Recommendation:

-Ensure that the data availability statement follows MDPI’s guidelines and includes a functional link.

Sensitivity Analysis: A sensitivity analysis would strengthen the study by evaluating how the model responds to variations in key factors.

Recommendations:

-Test different proportions of minority classes to proof of degree of imbalance.

-Experiment with different model configurations (e.g., GPR hyperparameters).

Author Response

Response to the reviewers’ comments

  “ Imbalanced Power Spectral Generation for Respiratory Rate and Uncertainty Estimations based on Photoplethysmography Signal  

Soojeong Lee, Mugahed A. Al-antari, Gyanendra Prasad Joshi, Yeong Hyeon Gu

Manuscript ID: Sensors-3427562

General

We appreciate the valuable comments and suggestions of the reviewers on our paper very much. We have incorporated all the reviewers’ comments and suggestions in our submitted manuscript and given additional explanations. Our detailed responses are as follows.

Reviewer #1:

  1. According to the comments,
  2. Abstract: While the abstract mentions improvements in accuracy and uncertainty management, it lacks a clear conclusion summarizing the impact or implications of the findings.

Recommendations:

-Include a concise conclusion at the end of the abstract that synthesizes the key findings and their significance in clinical enviroment.

  1. Answer.

We agreed with the reviewer's comments and included some sentences in the abstract as

 “Combining the proposed Gaussian process regression (GPR) with IPSG based on the Beth Israel Deaconess Medical Center dataset, the mean absolute error of RR is 0.79 and 1.47 brpm. Our approach achieves high stability and accuracy by randomly mixing original and artificial feature curves. The proposed GPR-IPSG model can improve the performance of clinical home-based monitoring systems and design a reliable framework.”

 Lines 20-25.

  1. According to the comments,

 Methods: The study appropriately employs relevant datasets (MIMIC-II and RRSYNTH) and methods (e.g., imbalanced feature generation using bootstrapping), aligning with its objective of evaluating a proposed algorithm for respiratory rate estimation. Signal processing steps and feature generation procedures are well-documented, demonstrating a clear intervention by the authors.

Recommendations: 

-Conduct a statistical significance analysis to determine whether the proposed method outperforms existing approaches. 

  1. Answer. We agree that a statistical significance analysis is critical to evaluate whether the proposed method outperforms existing approaches. Thus, we included the results of the ANOVA test as

We used to evaluate and compare the performance of the proposed IPSG with SVM, NN, GBA, LSTM, RF, and GPR algorithms based on the analysis of variance (ANOVA) test \cite{bailey} as shown in Tables 8 and 9.   ANOVA is a statistical method used in all situations requiring comparing two or more population means. That is, the hypothesis of interest in ANOVA is given $H_{0}: \mu_{1} = \mu_{2} ... = \mu_{j}$ and         $H_{1}: \mu_{1} \neq \mu_{2} ... \neq \mu_{j}$.

“The null hypothesis in ANOVA is that there is no difference in meaning. The alternative hypothesis is that the means are not all equal. Therefore, a multi-comparison was utilized to determine the group averages' results, which differed from the others.  One-way ANOVA is a simple and illustrative example of a linear model, given $e_{ij} = \alpha_{j} +\epsilon_{ij}$. Here, it was assumed that $e_{ij}$ was the experimental result (MAEs) of the proposed IPSG with SVM, NN, GBA, LSTM, RF, and GPR algorithms, where $i = 30$ was the number of measurements, and $j=6$ denoted the number of groups. 

Line 488-498  

“Based on the results of the ANOVA test on the left, as shown in Table \ref{tab8}, the performance of six ML algorithms was analyzed. Table \ref{tab8} presents the between-groups variation (Group) and within-groups variation (Errors), where SS is the sum of squares, and df is the degrees of freedom. The total degree of freedom is the total number of measurements (MAEs) minus one, which denotes 179 (=180-1). The between-groups degrees of freedom are the number of groups minus one, which denotes 5 (=6-1).  

The MS denotes the mean squared error (2.59), which denotes SS (12.96) /df (5). The F-statistic denotes the ratio of the mean squared errors (MS/Error). The $p$-value, 1.11 $\rm{e-44}$, denotes the probability that the test statistic can obtain a value greater than the value of the calculated test statistic, i.e., P(F > 149.26). The small $p$-value, 1.11 $\rm{e-44}$ < ($\alpha = 0.05$), denotes that differences between group means are significant. As shown in the right part of Table  \ref{tab8}, the $p$-value, 5.1 $\rm{e-46}$ is less than the significant value (0.05).  We confirm the results of the ANOVA test on the left as shown in Table \ref{tab9}, the very small $p$-value, 5.33 $\rm{e-61}$ < ($\alpha = 0.05$), represents that differences between group means are significant. The results of the ANOVA test on the right, as shown in Table \ref{tab9}, the $p$-value, 9.04 $\rm{e-67}$ < ($\alpha = 0.05$), denotes that differences between group means are significant.”

Line 591-606. 

  1. According to the comments,

Include a computational cost analysis and assess the feasibility of real-time implementation. 

  1. Answer. We included the sentences about computational cost analysis as

“The computational complexity of the GPR-IPSG model was similar to that of the traditional GPR model. Table 2 shows that the LSTM model consumed more computational resources than other ML models. Computational efficiency is an important factor in determining the practical applicability of a method, especially when deployed in real-time systems where speed and resource consumption are important. The key aspects of computational cost include time complexity, space complexity, and overall demand for processing power. However, the limitation of this study is that it focuses on the consumption of computer resources, how fast the six types of algorithms learn data when inputting experimental data, and how fast they estimate the target value (response variable) when inputting test data. According to the evaluation results, the LSTM-ISPG model consumed the most resources at 37.6 seconds (s) during model training, and GPR-ISPG consumed 1.92 s. The GBA model required 1.22 s for testing, and GPR-ISPG consumed 0.065 s. In addition, the data for training is 1402 continuous data samples and 350 simple samples for testing, which is expected to be sufficient for health monitoring in wearable smart devices.” 

Line 531-545. 

  1. According to the comments,

Ensure that the data availability statement follows MDPI’s guidelines and includes a functional link. 

  1. Answer. We included the data availability statement as

“Please refer to suggested Data Availability Statements in section ``MDPI Research Data Policies'' at http://peterhcharlton.github.io/RRest/bidmcdataset.html and  http://peterhcharlton.github.io/RRest/synthetic dataset.html.” 

5.According to the comments,  

Sensitivity Analysis: A sensitivity analysis would strengthen the study by evaluating how the model responds to variations in key factors.

Recommendations:

-Test different proportions of minority classes to proof of degree of imbalance.

  1. Answer. We included the sentence about the results of the test's different proportions to prove the degree of imbalance as

“Moreover, we obtained 120 samples from (7 $ <=$), 150 samples from (8 $>=$ and 9 $ <=$), 170 samples (10 $>=$ and 11 $ <=$), 160 samples from (22 $>=$ and 23 $ <=$),  130 samples from (24 $>=$ and 25 $ <=$), and acquired 100 samples from (26 $>=$) brpm samples from 636 original samples, used them as input data for the proposed IPSG algorithm, and acquired 836 artificial samples.  Thus, we acquired 1472 samples with 257 feature dimensions.  As with the synthetic data results above, in the third experimental scenario, we confirm how the results of the IPSG model and the proposed method were expressed when the ratio of artificial data was changed, as shown in Table \ref{tab6}.”

Line 462-469. 

“In the third experimental scenario, the results of the IPSG and the ML models when the changed ratio of artificial data can be seen in Table \ref{tab6}. The NN-IPSG method showed the best results in MAE, RMSE, and $R^{2}$. The remaining algorithms also showed excellent and stable results. The NN method showed many changes in the results due to adjusting the ratio of artificial synthetic data, so the NN algorithm may be considered sensitive.” 

Line 576-581. 

“In the second experimental scenario, the PPG signals were sequentially divided into (70\%) from as a training set and (30\%) from as a test set.   Due to space limitations in the paper, the results of the second experimental scenario  are denoted in Tables \ref{tab3}, \ref{tab4}, \ref{tab5} and \ref{tab6}.

We will skip the detailed analysis and the discussion. Overall, the results show a slight increase in  mean absolute error (MAE) and  root mean square error (RMSE)  compared to the first scenario experiment's results.” 

Line 354-359. 

6.According to the comments,

-Experiment with different model configurations (e.g., GPR hyperparameters).

  1. Answer. We included the sentence about the experiment with different model configurations as

“We show the experimental results using five kernel models based on the GPR algorithm, as shown in Table 5. Based on the results in Table 5, the GPR model denotes robust and excellent performance in various kernel environments.”

“In addition, we presented the results of MAE using five kernel models based on GPR and GPR-IPSG in the second experimental scenario, as shown in Table \ref{tab5}. The table shows that GPR and the proposed IPSG models are very stable and robust and show excellent results despite the various kernel changes, implying the high reliability of the GPR and IPSG algorithms.”

Line 572-576. 

Regards,

Soojeong Lee.

Reviewer 2 Report

Comments and Suggestions for Authors

Addressing the imbalance in samples between hypopnea and dyspnea using PPG signals is highly valuable in medicine. The authors propose applying bootstrap algorithms to generate imbalanced continuous features for accurately estimating the respiratory rate and subsequently compare the performance of several machine learning algorithms, which is a nice study that merits publication.

1. The key skill in this work is applying the GBA. However, I hope the authors could explain the issues of explaining the results. I also hope to understand how about the sensitivity of the concerned parameters.

2. In Table 5, the R square of the GBA IPSG model for BIDMC reaches 0.96, which is significantly higher than I expected. Could the authors please provide some explanations for this result? 

3. As seen in Fig. 9(b), the scattered data appears to contain a significant amount of regular patterns (approximate level line segment). Could there be an issue with the dataset? 

Author Response

Response to the reviewers’ comments 

 “ Imbalanced Power Spectral Generation for Respiratory Rate and Uncertainty Estimations based on Photoplethysmography Signal

Soojeong Lee, Mugahed A. Al-antari, Gyanendra Prasad Joshi, Yeong Hyeon Gu

Manuscript ID: Sensors-3427562

General

We appreciate the valuable comments and suggestions of the reviewers on our paper very much. We have incorporated all the reviewers’ comments and suggestions in our submitted manuscript and given additional explanations. Our detailed responses are as follows.

Reviewer #2: 

  1. According to the comments,

Addressing the imbalance in samples between hypopnea and dyspnea using PPG signals is highly valuable in medicine. The authors propose applying bootstrap algorithms to generate imbalanced continuous features for accurately estimating the respiratory rate and subsequently compare the performance of several machine learning algorithms, which is a nice study that merits publication.

1. The key skill in this work is applying the GBA. However, I hope the authors could explain the issues of explaining the results. I also hope to understand how about the sensitivity of the concerned parameters. 

1.Answer. We included the sentences about the results and parameters of the GBA as 

“The main parameters for each model were defined, and the possible range of values for each parameter was determined automatically using the optimized hyperparameters in MATLAB, highlighting our research’s potential benefits and advancements. The process began with conventional ML models like GBA and LSTM, which conducted a grid search for all possible parameter combinations to identify the parameters for achieving the best performance results.” 

Line 338-343. 

“As shown in Table 1, the GBA model shows stable results even with little parameter tuning and is robust to test data ratio adjustment.” 

“The GBA-IPSG model outperformed the conventional GBA model, with RMSE values of 1.73 brpm and 2.50 brpm, respectively. This difference indicates that the RMSE was reduced by 0.77 brpm. Therefore, a reduction of 0.77 brpm means that the performance improvement of RR is simply 44.5 \% = (2.50–1.73) / 1.73 × 100 with respect to the performance of the healthcare monitoring system. As shown in Table \ref{tab4}, the GBA model has a small number of parameters, shows stable results even with little parameter tuning, and shows robust results in the tuning range of the test data ratio from 20\% to 30\%.” 

Line 418-424.

“In the second experimental scenario, the PPG signals were sequentially divided into (70\%) from as a training set and (30\%) from as a test set.   Due to space limitations in the paper, the results of the second experimental scenario  are denoted in Tables \ref{tab3}, \ref{tab4}, \ref{tab5} and \ref{tab6}.

We will skip the detailed analysis and the discussion. Overall, the results show a slight increase in  mean absolute error (MAE) and  root mean square error (RMSE)  compared to the first scenario experiment's results.” 

Line 354-359. 

  1. According to the comments,
  1. In Table 5, the R square of the GBA IPSG model for BIDMC reaches 0.96, which is significantly higher than I expected. Could the authors please provide some explanations for this result? 
  1. Answer. We included the sentence concerning the GBA’s results as

“ The $\rm{R}^{2}$ (0.96) of the GBA-IPSG algorithm also denoted a stronger correlation with the response variables than the  $\rm{R}^{2}$ (0.57) of the GBA algorithm when using the BIDMC dataset with the first experimental scenario as shown in Figs. \ref{fig9}(e) and (f). These results imply that applying the proposed ISPG method in the hypopnea ($<$8brpm) subgroup improved performance, as shown in Table 7.” 

Line 612-616. 

“In addition, when IPSG was applied in dyspnea situations, the performance of ML was improved. GBA-IPSG (1.19 brpm) showed better results than GBA (1.94 brpm). NN-IPSG and GPR-ISPG also showed excellent results. In particular, applying the IPSG method to all ML algorithms in hypopnea ($<$ 8 brpm) situations showed much better results than when only the ML algorithm was used. These results indicate surprising changes when clinical data are sparse, such as hypopnea ($<$ 8 brpm).” 

Line 582-590. 

  1. According to the comments,
  1. As seen in Fig. 9(b), the scattered data appears to contain a significant amount of regular patterns (approximate level line segment). Could there be an issue with the dataset? 
  1. Answer. We included a detailed sentence about the Fig. 9(b) as

“Figure 9(b) shows the proposed IPSG artificial data result as a horizontal pattern. This is the artificial label for IPSG estimation.” 

Line 512-514. 

Regards,

Soojeong Lee.

Reviewer 3 Report

Comments and Suggestions for Authors

Summary:

Machine learning algorithms can be impacted due to imbalance across different sub-groups of respiration: normal (12–20 brpm), dyspnea (≥20 brpm), and hypopnea (<8 brpm). The authors propose imbalanced power spectral generation using bootstrap to estimate respiratory rate and uncertainty based on the photoplethysmogram to address this issue.

General concept comments:  

While the work is scientifically sound with appropriate study design, there are following limitations in the study which needs to be addressed prior to publication. 

In Figure 2b and 2d, the maximum respiratory rate in the extended distribution seems to be around 25brpm. This may not be sufficient as tachypnea can be as high as 45brpm in adults and in children it can be much higher. Please clearly indicate the range of respiratory rate in the figure or the text. If the database does not contain respiratory rates higher than 25 brpm, please discuss it as a limitation of the study.

One of the key performance metrics used in the study is MAE. However, it is not clear how the MAE varies across different subgroups of respiration. Please include a subgroup analysis of performance for normal (12–20 brpm), dyspnea (≥20 brpm), and hypopnea (<8 brpm). 

The authors have provided box plots, regression plots and CI values. For completeness, please provide and discuss Bland-Altman analysis with 95% limits of agreement.

Minor comments:

Please use breaths per minute (brpm) when referring to respiratory rate. Beats per minute (bpm) can be used to refer to heart rate.

Line 361 to 365 is not clear. Please revise.

As these comments are straightforward to address, I recommend accepting the article after the comments are satisfactorily addressed.

Author Response

Response to the reviewers’ comments 

 “ Imbalanced Power Spectral Generation for Respiratory Rate and Uncertainty Estimations based on Photoplethysmography Signal 

Soojeong Lee, Mugahed A. Al-antari, Gyanendra Prasad Joshi, Yeong Hyeon Gu

Manuscript ID: Sensors-3427562

General

We appreciate the valuable comments and suggestions of the reviewers on our paper very much. We have incorporated all the reviewers’ comments and suggestions in our submitted manuscript and given additional explanations. Our detailed responses are as follows.

Reviewer #3:

1.According to the comments,  

In Figure 2b and 2d, the maximum respiratory rate in the extended distribution seems to be around 25brpm. This may not be sufficient as tachypnea can be as high as 45brpm in adults and in children it can be much higher. Please clearly indicate the range of respiratory rate in the figure or the text. If the database does not contain respiratory rates higher than 25 brpm, please discuss it as a limitation of the study.

1.Answer. We added explanations about our database’s limitation as 

“Although the BIDMC data does not contain respiration rates above 25 brpm, which is a limitation of the BIDMC data set, we completed the experiment using the RRSYNTH data set. Although the RRSYNTH data also completed the experiment by generating artificial data up to about 25 brpm, we plan to experiment by generating artificial data above 30 brpm in future studies.” 

Line 692-696. 

2.According to the comments,

One of the key performance metrics used in the study is MAE. However, it is not clear how the MAE varies across different subgroups of respiration. Please include a subgroup analysis of performance for normal (12–20 brpm), dyspnea (≥20 brpm), and hypopnea (<8 brpm). 

  1. Answer. We included the results of MAE varies different subgroup of respiration for normal (12-20 brpm), dyspnea (>=brpm), hypopnea (<brpm) in Table 7 as

“Additionally, we include the results of subgroups MAE for normal (12–20 brpm), dyspnea ($\geq20$brpm), and hypopnea ($<$8 brpm) as shown in Table \ref{tab7}. These results imply that applying the proposed IPSG method in the hypopnea ($<$8brpm) subgroup improved performance, as shown in Table 7.” 

Line 484-487. 

“The MAE results as denoted in Table \ref{tab7}, we found that in normal breathing situations (12-20 brpm), the results of MAE were almost similar or that the performance of ML decreased when IPSG was applied. In addition, when IPSG was applied in dyspnea situations, the performance of ML was improved. GBA-IPSG (1.19 brpm) showed better results than GBA (1.94 brpm). NN-IPSG and GPR-ISPG also showed excellent results. In particular, applying the IPSG method to all ML algorithms in hypopnea ($<$ 8 brpm) situations showed much better results than when only the ML algorithm was used. These results indicate surprising changes when clinical data are sparse, such as hypopnea ($<$ 8 brpm).” 

Line 582-589. 

3.According to the comments,

The authors have provided box plots, regression plots and CI values. For completeness, please provide and discuss Bland-Altman analysis with 95% limits of agreement.

  1. Answer. We included the sentence about Bland-Altman plot as

“The GPR and GPR-ISPG models, differed minimally from the reference RR values. The agreement limits are indicated by the red horizontal lines in Figs. \ref{fig11}(a) and (b) are  mean error (ME) $\pm$2$\times$SD.  Fig. \ref{fig11}(a) shows the Bland-Altman plot of reference RR, with the ME of 0.13 mmHg and the SD $\pm 1.96$ brpm. Fig. \ref{fig11}(b) presents the Bland-Altman plot of reference RR, with the ME of -0.03 brpm and the SD $\pm 1.40$ brpm.

The SDs of RF and RF-ISPG are clustered tightly within a narrow range compared to the baseline RR. The reliability of the RF and RF-ISPG algorithms is highlighted in Figures \ref{fig11}(c) and (d), which compare the SDs ($\pm 2.46$ brpm and $\pm 1.82$ brpm) of the MEs (0.19 brpm and 0.08 brpm) of RF and RF-ISPG with the baseline RR. The SDs of RF and RF-ISPG show consistent and reliable performance compared to the baseline RR, which ensures their performance.

The Brand-Artman plots of GBA and GBA-IPSG in Figs \ref{fig11}(e) and (f) also show that they are located within the small MEs (-0.23 brpm and -0.18 brpm) and narrow SDs ($\pm 2.76$ brpm and $\pm 1.84$ brpm) compared to the baseline RR. These results also imply reliable results. These results confirm that GPR-ISPG, RF-ISPG, and GBA ISPG agree better with the reference RR than GPR, RF, and GBA. This is because the ME of the estimated RR using the proposed ISPG methodology is close to 0, and most of the blue and black points are within the upper and lower bounds.” 

Line 645-661.

4.According to the comments,

Please use (brpm) when referring to respiratory rate. Beats per minute (bpm) can be used to refer to heart rate.

  1. Answer. In this manuscript, the abbreviation bpm is modified to brpm.

“The sample differences between normal breathing (12–20 breaths per minute (brpm)), dyspnea ($\geq$20 brpm), and hypopnea ($<$8 brpm) show significant data imbalance, which can affect the learning of machine learning algorithms.” 

5.According to the comments,

  1. Line 361 to 365 is not clear. Please revise.
  2. Answer. We revised the sentence as

“Furthermore, we obtained 100 samples from (7 $ <=$),  100 samples from (8 to 9), 100 samples from (10 to 11), 100 samples from (22 to 23), 100 samples (24 to 25), and acquired from 100 samples from (26 $>=$ brpm) using 636 original samples, used them as input data for the proposed IPSG algorithm, and acquired 600 artificial samples using the proposed IPSG method as denoted in Fig. \ref{fig4}(b).”

Line 368-372 

Regards,

Soojeong Lee.